# *Trans*-eQTL mapping prioritises *USP18* as a negative regulator of interferon response at a lupus risk locus

Krista Freimann[1,2], Anneke Brümmer [3,4], Robert Warmerdam [5,6], Tarran S. Rupall[2,7], Ana Laura Hernández-Ledesma [8], Joshua Chiou [9], Emily R. Holzinger[10], Joseph C. Maranville [10], Nikolina Nakic[11], Halit Ongen[12], Luca Stefanucci [2,7], Michael C. Turchin [10], eQTLGen Consortium*, Lude Franke [5,6], Urmo Võsa [13], Carla P. Jones [2,7], Alejandra Medina-Rivera[8], Gosia Trynka [2,7], Kai Kisand [14], Sven Bergmann [3,4,15] & Kaur Alasoo [1,2] ✉

Although genome-wide association studies have provided valuable insights into the genetic basis of complex traits and diseases, translating these findings to causal genes and their downstream mechanisms remains challenging. We performed *trans* expression quantitative trait locus (*trans*-eQTL) meta-analysis in 3734 lymphoblastoid cell line samples, identifying four robust loci that replicated in an independent multi-ethnic dataset of 682 individuals. The *trans*-eQTL signal at the ubiquitin specific peptidase 18 (*USP18)* locus colocalised with a GWAS signal for systemic lupus erythematosus (SLE). USP18 is a known negative regulator of interferon signalling and the SLE risk allele increased the expression of 50 interferon-inducible genes, suggesting that the risk allele impairs USP18's ability to effectively limit the interferon response. Intriguingly, the *USP18 trans*-eQTL signal would not have been discovered in a meta-analysis of up to 43,301 whole blood samples, reaffirming the importance of capturing context-specific genetic effects for GWAS interpretation.

Genome-wide association studies (GWAS) have provided valuable insights into the genetic basis of complex traits and diseases. However, translating GWAS findings to actionable drug targets has remained challenging, particularly when the functions of the associated genes are unknown. A promising technique to identify the effector genes of GWAS variants as well as their downstream

regulatory consequences is *trans* gene expression and protein quantitative trait loci (*trans*-QTL) analysis. *Trans*-QTL studies test for association between genetic variants across the genome and expression levels of all measured genes or proteins[1]. In a prominent example, an erythrocyte-specific regulatory element first identified as a *trans* protein QTL (*trans*-pQTL) for foetal haemoglobin (HbF)

[1]Institute of Computer Science, University of Tartu, Tartu, Estonia. [2]Open Targets, South Building, Wellcome Genome Campus, Hinxton, Cambridge, UK. [3]Department of Computational Biology, University of Lausanne, Lausanne, Switzerland. [4]Swiss Institute of Bioinformatics, Lausanne, Switzerland. [5]Department of Genetics, University of Groningen, University Medical Center Groningen, Groningen, the Netherlands. [6]Oncode Institute, Amsterdam, the Netherlands. [7]Wellcome Sanger Institute, Wellcome Genome Campus, Hinxton, UK. [8]Laboratorio Internacional de Investigación Sobre el Genoma Humano, Universidad Nacional Autónoma de, México, Santiago de Querétaro, Mexico. [9]Internal Medicine Research Unit, Research and Development, Pfizer, Cambridge, MA, USA. [10]Bristol Myers Squibb, Massachusetts, MA, USA. [11]Research Technologies, GSK, Stevenage, UK. [12]Research Technologies, GSK, Heidelberg, Germany. [13]Estonian Genome Centre, Institute of Genomics, University of Tartu, Tartu, Estonia. [14]Institute of Biomedicine and Translational Medicine, Faculty of Medicine, University of Tartu, Tartu, Estonia. [15]Department of Integrative Biomedical Sciences, University of Cape Town, Cape Town, South Africa. *A list of authors and their affiliations appears at the end of the paper. ✉e-mail: kaur.alasoo@ut.ee

was used to design the first ever gene editing therapy for sickle-cell disease[2,3].

*Trans*-QTLs are especially promising, because 60%-90% of gene and protein expression heritability is located in *trans*[4], most associations detected in large-scale pQTL studies are located in *trans*[5], and *cis*-QTL discovery is starting to saturate after 10,000 samples[5]. Furthermore, most complex trait heritability has been proposed to be mediated by *trans*-QTL effects[4]. However, current large-scale *trans*-eQTL and *trans*-pQTL studies have been limited to easily accessible bulk tissues such as whole blood[6,7] or plasma[5,8–10]. Bulk tissue studies are subject to cell type composition effects which can be difficult to distinguish from true intracellular *trans*-QTLs[1,6]. The whole blood and plasma studies are also likely to miss cell type and context specific regulatory effects. In contrast, *trans*-eQTL studies in other tissues and purified cell types have had limited statistical power due to small sample sizes (typically less than one thousand samples), enabling the discovery of only very large effects and potentially underestimating pleiotropic effects on multiple target genes[11–18].

A key limitation in our understanding of how *trans*-eQTLs contribute to complex traits and how they interact with *cis*-eQTL is the lack of well-characterised disease-associated *trans*-eQTL signals[4]. Two most prominent examples include the adipose-specific *KLF14* locus associated with type 2 diabetes[16,19] and the IRX3/5 locus associated with obesity[20,21]. At the *KLF14* locus, the lead variant (rs4731702) is a *cis*-eQTL for the *KLF14* transcription factor and was associated with the expression of 385 target genes in *trans*, 18 of which also had independent *cis* associations for other metabolic traits[19]. The simultaneous regulation of multiple target genes in *trans*-eQTL regulatory networks seems to be a general property of many known *trans*-eQTL signals[6,12,14]. However, what proportion of *trans*-eQTL target genes directly mediate the disease or trait associations as opposed to being independent 'bystanders' with minimal direct causal effect has remained unclear.

We performed the largest *trans*-eQTL meta-analysis in a single cell type, comprising 3734 lymphoblastoid cell line (LCL) samples across nine cohorts (MetaLCL). LCLs are obtained by transforming primary B-cells with Epstein-Barr virus[22]. LCLs have been widely used as a resource for human genetics, from banking cells from rare genetic disorders, through control material in laboratories to prevent repetitive blood sampling, to the study of tumorigenesis, mechanisms of viral latency and immune evasion[22]. Furthermore, Epstein-Barr virus has been epidemiologically linked to several autoimmune diseases in which B cells are implicated to play a pathogenic role, such as multiple sclerosis (MS)[23,24] and systemic lupus erythematosus (SLE)[25] with recent studies starting to elucidate the potential molecular mechanisms underlying these associations[26–28]. Thus, *trans*-eQTLs discovered in LCLs might provide insights into the pathogenesis of these autoimmune diseases, especially in the context of chronic exposure to viral stimuli.

After stringent quality control, we identified four highly robust *trans*-eQTL associations that replicated in an independent cohort (*n* = 682) and were associated with multiple target genes. At the *USP18* locus, the *trans*-eQTL signal colocalised with a GWAS association for SLE. The SLE risk allele was associated with increased activity of the type I interferon signalling pathway and increased expression of several classical interferon response genes. While there is robust evidence for the potential causal role of increased interferon signalling in SLE pathogenesis, we find that the expression of many individual interferon response genes is unlikely to have a direct causal effect on SLE. Our results caution against blindly using *trans*-QTL associations for target gene prioritisation without clear understanding of the *trans*-QTL mechanism and robust genetic evidence from *cis*-acting variants implicating the same gene. To support secondary use of our data, we have made the complete MetaLCL *cis* and *trans*-eQTL summary statistics for 18,792 genes publicly available via the eQTL Catalogue FTP server.

## Results

### Large-scale *trans*-eQTL meta-analysis in a single cell type

We performed a large-scale *trans*-eQTL study, utilising data from LCLs collected from 3734 donors (2238 female, 1496 male) across nine cohorts of European ancestries (Supplementary Table 1). To avoid confounding by technical factors, we performed association testing separately in each cohort and meta-analysed the results (Fig. 1A). After excluding *cis* associations located within 5 Mb of the target gene, we identified 79 suggestive independent *trans*-eQTL loci at $p < 1 \times 10^{-11}$ threshold (Fig. 1B). To identify robust signals associated with multiple target genes and reduce the risk of false positives caused by cross-mappability[29], we further required each locus to be associated with at least five independent target genes ($p < 5 \times 10^{-8}$) with low cross-mappability scores (see Methods). This filtering reduced the number of candidate loci to six (Fig. 1B), four of which replicated in an independent multi-ethnic cohort of 682 individuals[30]. These four replicating *trans*-eQTL loci were located near the *BATF3*, *MYBL2*, *USP18*, *HNF4G* genes (Supplementary Data 1, Supplementary Fig. 2). While the strong *trans*-eQTL signal near the *BATF3* transcription factor (2294 targets at FDR 5%) has been previously reported[31], the other three seem to be novel. Remarkably, the *trans*-eQTL targets at the *MYBL2* locus were consistent with direct activation by the MYBL2 transcription factor (Supplementary Note 1), indicating that our analysis is identifying biologically interpretable signals.

### *USP18* is a negative regulator of interferon response at a lupus GWAS locus

To prioritise the four *trans*-eQTL loci for follow-up analysis, we performed GWAS lookup using the Open Targets Genetics Portal[32] and found that only the *USP18* locus lead variant (chr22_18166589_T_C, rs4819670) was in high LD ($r^2 > 0.9$) with an annotated GWAS hit. Specifically, the *USP18* lead variant was identical to a GWAS lead variant reported for SLE in East Asians[33]. Using the point estimation of colocalisation (POEMColoc) method, we confirmed that the two signals colocalised (PP4 = 0.97) (Fig. 2A)[34]. The colocalisation also replicated in an SLE GWAS meta-analysis across the UK Biobank (UKBB)[35], FinnGen[36] and Million Veterans Program (MVP)[37] biobanks (Supplementary Fig. 3)[38]. At this locus, we identified 40 *trans* target genes at false discovery rate (FDR) 5% that were all strongly enriched for Reactome interferon signalling (R-HSA-913531, $p = 1.1 \times 10^{-26}$) and interferon alpha/beta signalling (R-HSA-909733, $p = 1.7 \times 10^{-21}$) pathways. The rs4819670-C allele was associated with decreased expression of multiple canonical type I interferon response genes (e.g., *ISG15*, *IFI44*, *OAS1-3*) (Fig. 2B). Reassuringly, we observed consistent effect sizes across nine sub-cohorts in our meta-analysis ($I^2$ heterogeneity statistic = 0.46, Supplementary Fig. 2, Supplementary Data 2). The rs4819670-C was also associated with decreased risk of systemic lupus erythematosus (SLE) in East Asians[33] as well as in the MVP-FinnGen-UKBB meta-analysis[38].

We next sought to identify the most likely causal gene at the *USP18* locus. Since the *trans*-eQTL effect was detected in monocultures of LCLs[39], this implies that the causal *cis* effect mediating the *trans* associations must also be present in the same cell type (i.e., it cannot be mediated by a *trans*-acting factor such as a cytokine produced by some other cell type). Although the rs4819670 variant was located in an intron of *USP18*, we did not detect a significant *cis*-eQTL effect for *USP18* or any other neighbouring gene neither in our meta-analysis nor the full eQTL Catalogue release 6[40]. We also did not detect a splicing QTL for *USP18* or other neighbouring genes in eQTL Catalogue release 6. However, the rs4819670 lead variant was in high linkage disequilibrium (LD) with a *USP18* missense variant rs3180408 (chr22_18167915_C_T, ENSP00000215794.7:p.Thr169Met) in both European and East Asian populations (r = −1, $r^2 = 1$ in EAS and r = −0.98, $r^2 = 0.96$ in EUR 1000 Genomes superpopulation). Furthermore, the rs3180408 missense variant was the lead variant for SLE in the

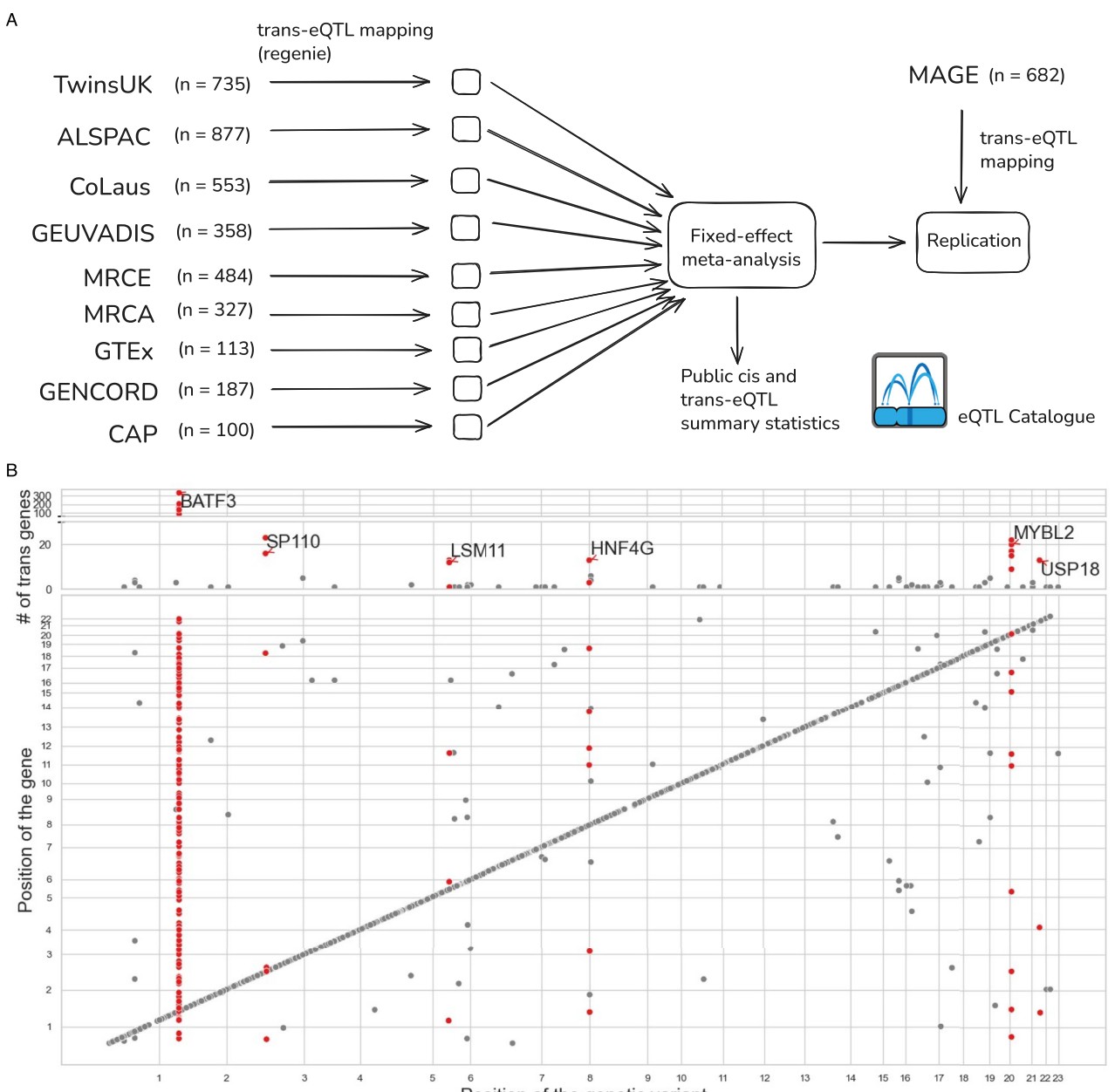

**Fig. 1 | Overview of the MetaLCL study. A** Overview of the study design and participating cohorts. **B** Significant *trans*-eQTLs detected in the meta-analysis. The upper scatter plot shows the number of *trans*-eQTL target genes detected at each *trans*-eQTL locus with $p < 5 \times 10^{-8}$. Six *trans*-eQTL loci with the most target genes have been labelled with the name of the closest *cis* gene. The lower scatter plot shows all significant loci for each tested gene at the more stringent $p < 1 \times 10^{-11}$ threshold. *Cis* associations are located on the diagonal while putative *trans* associations are located off diagonal. The points represent two-sided *p*-values from inverse-variance weighted meta-analysis. Source data are provided as a Source Data file.

MVP-FinnGen-UKBB meta-analysis (Supplementary Fig. 3). Notably, *USP18* is a known negative regulator of interferon signalling and a rare loss-of-function mutation in *USP18* causes severe type I interferono-pathy (Fig. 2C)[41,42]. Similarly, *USP18* knock-out in human macrophages increased the expression of several canonical interferon response genes upon stimulation with interferon-beta[43]. Thus, these results indicate that *USP18* is the most likely causal gene at this *trans*-eQTL locus.

However, identifying the exact causal variant remains challenging. While the absence of *cis* eQTL and splicing QTL evidence suggests that the likely causal mechanism is the rs3180408 missense variant, the missense variant was predicted to be benign by all tested variant effect prediction tools available from Ensembl VEP[44]. Alternatively, there

could be other genetic variants in the region that are not captured by current genotype imputation reference panels (such as structural variants). One potential strategy to assess the functional impact of the rs3180408 missense variant on *trans*-eQTL target gene expression would be genome editing, but our power calculations (see Supplementary Note 2) suggest that this would be extremely challenging due to the small expected effect size of the variant.

**Role of aberrant interferon signalling in lupus pathogenesis**
Several studies have suggested that causal GWAS genes are enriched in shared pathways or biological processes[45-47]. To further characterise the potential role of USP18 target genes in lupus, we performed additional *trans*-eQTL meta-analysis across the nine discovery cohorts

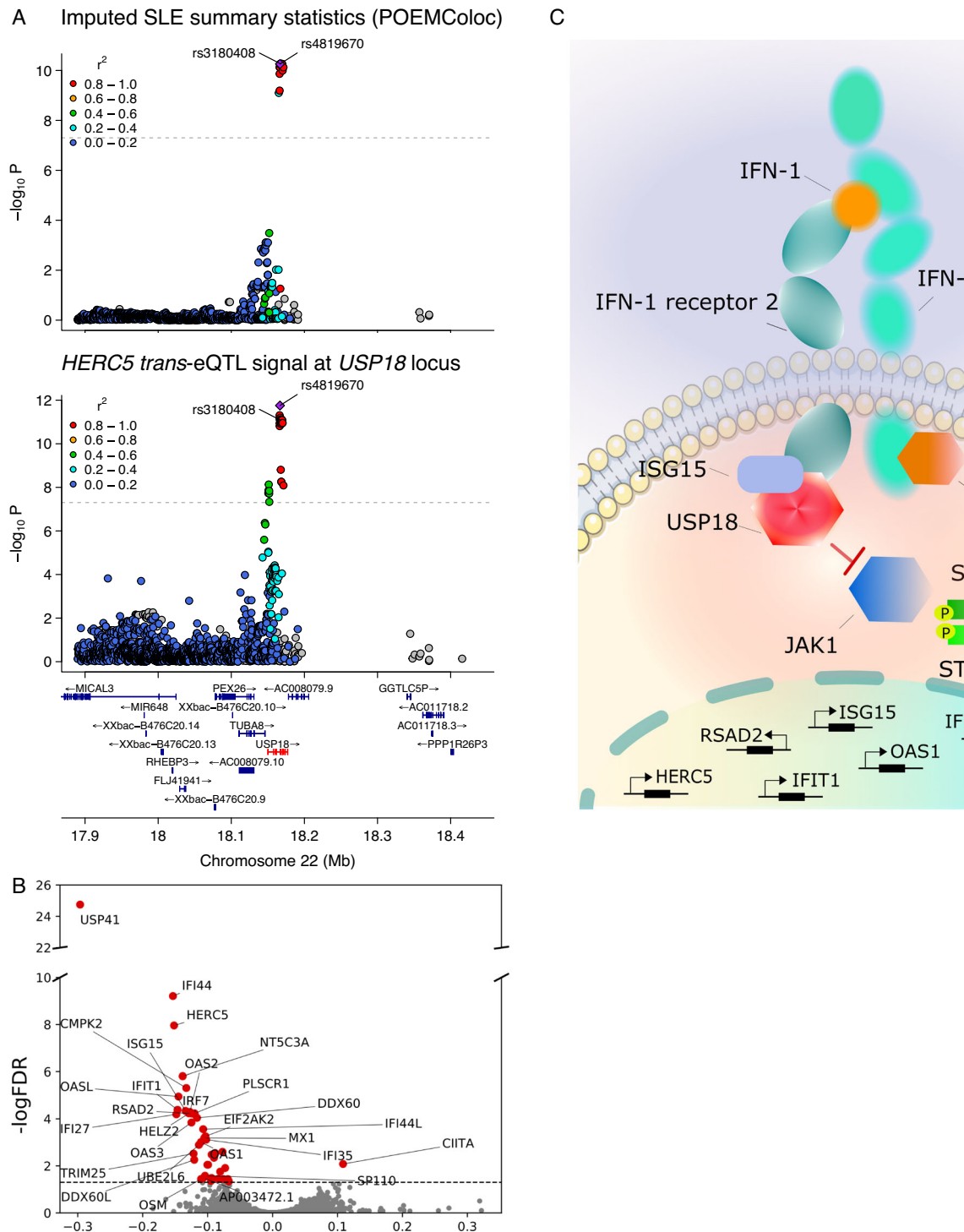

**Fig. 2 | SLE GWAS association at the *USP18* locus is a *trans*-eQTL for interferon response genes. A** Regional association plot for the SLE GWAS with POEMColoc imputed summary statistics and regional association plot for the lead *trans*-eQTL gene (*HERC5*) at the *USP18* locus. The *trans*-eQTL lead and GWAS lead variants (rs4819670) are identical and in high LD with a missense variant (rs3180408) in the *USP18* gene. The points represent two-sided *p*-values from inverse-variance weighted meta-analysis. The original regional association plot for the SLE GWAS is shown on Supplementary Fig. 4. **B** Volcano plot of the *trans*-eQTL target genes. Genes with FDR < 5% are highlighted in red. **C** USP18 down-regulates type I interferon signalling by restricting the access of Janus-associated kinase 1 (JAK1) to the type I interferon receptor[42]. Source data are provided as a Source Data file.

and one replication cohort (total *n* = 4416). This increased the number of significant USP18 target genes to 50 (FDR < 5%). Notably, 18/50 target genes overlapped the Reactome interferon alpha/beta signalling (R-HSA-909733) pathway (hypergeometric test, *p* = 4.14 × 10⁻²⁴) and

26/50 genes overlapped a consensus set of interferon response genes (*n* = 124) identified by Mostafavi et al.[48] (*p* = 1.44 × 10⁻³⁹, Supplementary Data 3). Reassuringly, 40/50 genes were also more highly expressed in peripheral blood mononuclear cells from SLE cases compared to

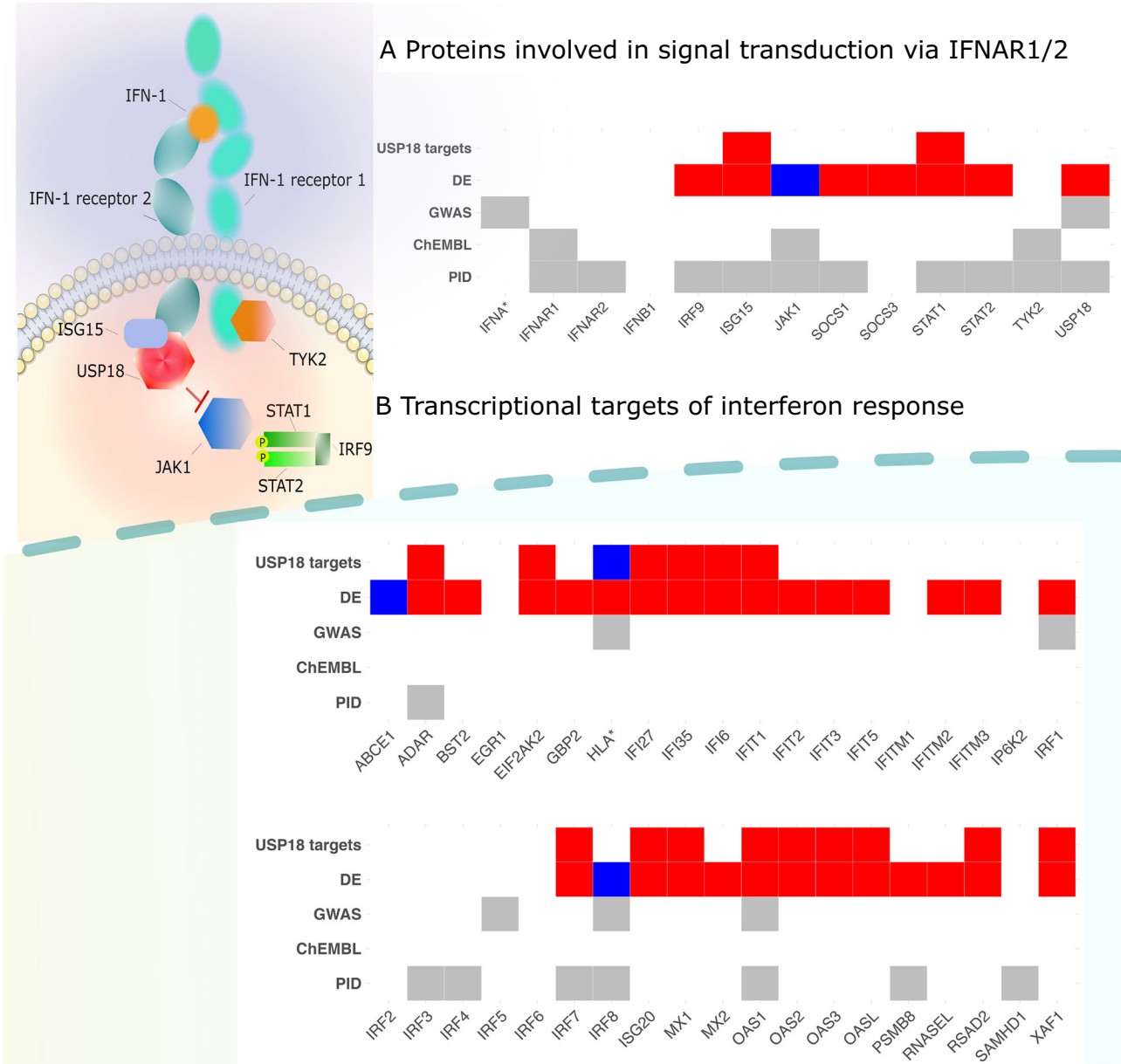

**Fig. 3 | Role of interferon signalling in SLE pathogenesis. A** Upstream regulators of interferon response genes (IFNA* contains multi-gene interferon-alpha gene cluster). **B** Downstream transcriptional targets of the interferon signalling (HLA* marks the HLA region). The increased gene expression is marked in red, while reduced gene expression is marked in blue. The visualisation illustrates the effect on USP18 target genes in relation to the SLE risk allele. DE - differential gene expression in SLE cases *versus* controls[49]; GWAS - GWAS hits for SLE[33], ChEMBL, phase III - SLE phase III clinical trials from ChEMBL[56], PID - genes causing primary immunodeficiency from Genomics England. Source data are provided as a Source Data file.

controls[49] (Supplementary Data 3), consistent with the established role of increased interferon signalling in SLE[50].

To better understand the role of the USP18 target genes in the interferon alpha/beta signalling pathway, we focussed on the 60 genes belonging to the Reactome R-HSA-909733 interferon alpha/beta signalling pathway and divided them into three categories - category I: proteins involved in signal transduction via IFNAR1/2 receptor (*n* = 13 genes, including the multi-gene interferon-alpha gene cluster, Fig. 3A); category II: downstream transcriptional targets of the interferon signalling (38 genes from the Reactome R-HSA-1015702 sub-pathway, Fig. 3B) and category III: other pathway genes (*n* = 9) not belonging to the first two categories (Supplementary Fig. 5). We found that 16/50 USP18 targets were shared with the 38 category II genes (transcriptional targets of interferon response) (*p* = 1.18 × 10⁻²⁹, Fisher's exact test). In contrast, only 2/50 USP18 *trans*-eQTL target genes (*STAT1* and *ISG15*) were shared with the 13 category I genes (IFNAR1/2 receptor signal transduction proteins) and none were shared with the 9 category III genes. This suggests that the USP18 *trans*-eQTLs are primarily capturing the transcriptional targets of interferon response (category II), consistent with the established role of USP18 in regulating these genes (Fig. 2C)[42].

Next, we assessed if there were additional lupus GWAS signals overlapping the three categories of interferon response genes defined above. We first used the Open Targets Genetics portal to extract the prioritised target genes for 108 lupus GWAS loci from Yin et al.[33]. This revealed that three prioritised lupus genes (*USP18, STAT1, IFNA1-17*) were shared with the 13 category I genes (IFNAR1/2 signal transduction proteins, Fig. 3A) and four prioritised genes (*IRF1/5/8* and *OAS1*) were

shared with the 38 category II genes (transcriptional targets of interferon response, Fig. 3B). Out of these, IRF1/5/8 are themselves transcription factors involved in the regulation of interferon production[51,52], and stronger IRF1 binding across many GWAS loci has been associated with higher Crohn's disease risk[53]. *OAS1* represents a classical antiviral gene and here the SLE GWAS lead variant is in perfect LD with a fine-mapped splice QTL for *OAS1* in the eQTL Catalogue (Supplementary Fig. 6)[40]. The consequences of this splice variant have been extensively studied in the context of Sjögren's syndrome and COVID-19 hospitalisation[54,55]. Unfortunately, we were not able to check if there were additional sub-threshold hits for lupus near the interferon alpha/beta signalling pathway genes, because the genome-wide summary statistics from Yin et al.[33] were not available.

We also overlapped interferon alpha/beta signalling pathway genes with ongoing or completed phase III clinical trials for SLE extracted from the ChEMBL database[56]. We identified three category I (interferon signal transduction) genes (*IFNAR1*, *JAK1* and *TYK2*) that have been targeted by a clinical trial for SLE (Fig. 3A). While the trials targeting *JAK1* and *TYK2* are currently ongoing, a randomised control trial of anifrolumab, a human monoclonal antibody to type I interferon receptor subunit 1 (*IFNAR1*), found it to be an effective treatment for SLE[57]. None of the category II genes (transcriptional targets of interferon signalling, Fig. 3B) and category III genes (Supplementary Fig. 5) are currently in a phase III clinical trial for SLE (Fig. 3B).

There is an emerging consensus that rare mutations in genes prioritised for autoimmune diseases from GWAS studies can often also cause primary immunodeficiencies (PIDs)[58–60]. For example, loss-of-function mutations in *USP18* cause rare type I interferonopathy[41,42]. At the same time, GWAS studies for SLE and other autoimmune diseases are still only powered to detect variants with large effects. Thus, knowing if a gene causes PID might be a useful (if noisy) indicator that the same gene might be discovered in a future larger autoimmune GWAS study. We obtained the list of genes causing either PID or monogenic inflammatory bowel disease from Genomics England[61] and overlapped those with the three categories of interferon response genes defined above. We found that 10/13 category I genes (interferon signal transduction) have previously been implicated in causing PID, including *USP18* and all three phase III drug candidates for SLE (Fig. 3A). In contrast, only 8 of the 38 category II genes (transcriptional targets of interferon response) have been implicated in PIDs (Fig. 3B), including *OAS1* and *IRF8* also detected by SLE GWAS. Finally, none of the category III genes have been implicated in PIDs (Supplementary Fig. 5).

Triangulation of evidence from prioritised lupus GWAS target genes, phase III clinical trial information and overlap with primary immunodeficiency genes reaffirms modulation of aberrant interferon alpha/beta signalling in B-cells as an emerging therapeutic opportunity for SLE (category I, Fig. 3A)[60,62]. This is further supported by recent studies demonstrating that depleting autoreactive B-cells via anti-CD19 CAR T cell therapy is an effective therapy for SLE and other autoimmune diseases[63,64]. In contrast, most *trans*-eQTL targets of USP18 are transcriptional targets of interferon response (category II, Fig. 3B) and it is far less clear what are the potential causal roles of these genes in SLE pathogenesis.

### Replication of the *USP18 trans*-eQTL signal in whole blood
To understand the context-specificity of the *USP18 trans*-eQTL signal, we performed additional replication in the eQTLGen Phase 2 *trans*-eQTL meta-analysis of up to 43,301 whole blood samples. We observed that the *USP18* missense variant rs3180408 was nominally associated ($p < 0.05$) with the expression of 7/50 USP18 target genes, including our lead target gene *HERC5* ($p = 0.037$) as well as canonical interferon response genes *IFI44* and *ISG15* (Supplementary Data 4). For 6/7 nominally significant associations, the effect direction was concordant between the LCL and whole blood meta-analyses, but the effect size

was an order of magnitude smaller in whole blood (Supplementary Data 4). We further tried to replicate the *USP18 trans*-eQTL association in naive B cells using single-cell RNA-seq data from 844 individuals from the OneK1K cohort[18], but none of the tested genes were associated with the *trans*-eQTL lead variant (Supplementary Data 5). Thus, even at these large sample sizes, the *USP18 trans*-eQTL signal would not have been discovered in whole blood or naive B-cells.

To understand the potential reasons for the attenuated effect in whole blood, we compared the expression level of the *USP18* gene across 49 GTEx tissues. We found that *USP18* had the highest expression in LCLs (median transcripts per million (TPM) = 45.3) and one of the lowest in whole blood (median TPM = 0.46). Since *USP18* is itself an interferon response gene and LCLs are characterised by a strong interferon signature driven by active infection with the Epstein-Barr virus, we characterised the expression of *USP18* in naive B-cells as well as B-cells stimulated with interferon-alpha and TLR7/8 agonist R848 for 16, 40 and 64 h. We found that the expression level of *USP18* in B-cells was upregulated by ~3.5-fold after 16 h of stimulation and stayed elevated for at least 64 h (Supplementary Fig. 7). This suggests that the very strong active interferon signalling and associated upregulation of *USP18* transcription in LCLs is required for the *trans*-eQTL signal to be detected.

## Discussion
We performed the largest *trans*-eQTL study in a single cell type where we profiled the expression of 18,792 genes in 3737 individuals from nine cohorts. We then replicated these findings in an independent multi-ancestry LCL cohort of 682 individuals. After careful quality control, we identified six independent loci that were associated with five or more target genes, and that were unlikely to be driven by cross-mappability artefacts. While we primarily focussed on the SLE-associated *USP18* locus in our analysis, we have publicly released the complete genome-wide summary statistics from our MetaLCL project via the eQTL Catalogue FTP server. In addition to disease-specific colocalisation applications, we expect that our summary statistics will motivate the development and application of novel summary-based aggregative *trans*-eQTL mapping methods[65–67].

While the GWAS signal for SLE at the *USP18* locus was discovered in East Asian cohorts[33], our *trans*-eQTL analysis was based on samples of predominantly European genetic ancestry. The shared GWAS and *trans*-eQTL lead variant (rs4819670) is common in both genetic ancestry groups (EAS MAF = 0.16; EUR MAF = 0.36), indicating that allele frequency differences alone cannot explain why the SLE GWAS association has not been detected in European ancestry individuals. However, prevalence, severity and age of onset of SLE varies considerably between ancestry groups[68]. Furthermore, heterogeneity of GWAS effect sizes between European and East Asian SLE GWAS studies has been reported[68]. Mechanisms of this genetic heterogeneity are unknown, but one plausible explanation is the presence of gene-environment interactions (e.g., diet or exposure to regional pathogens). In contrast, eQTL effect sizes in cultured LCLs are highly concordant between genetic ancestry groups[30], indicating that our use of European-ancestry samples to interpret a GWAS hit discovered in East Asians is unlikely to systematically bias our results in this case. Furthermore, the *USP18 trans*-eQTL signal replicated in a multi-ancestry cohort[30], indicating that the *trans*-eQTL effect is shared across ancestries. Finally, the *USP18* SLE GWAS signal is also present in the MVP-FinnGen-UKBB meta-analysis (Supplementary Fig. 3).

Despite the strong evidence for the critical role of type I interferon response in SLE pathogenesis[49,50] and three active clinical trials, we were surprised to see that of the 50 USP18 target genes, only *OAS1* had an independent *cis*-association with SLE. Expanding the analysis to interferon response genes from Reactome further implicated IRF1/5/8 genes and the HLA region, but most interferon response genes were not detected in the SLE GWAS. One potential explanation for this could

be the limited statistical power of the SLE GWAS that profiled 13,377 cases and 194,993 controls, identifying a total of 113 loci[33]. Furthermore, Liu et al. demonstrated that if multiple effector genes ('core' genes) are co-regulated by shared *trans* factors, with shared directions of effects (which seems to be the case for the interferon response genes), then nearly all heritability would be due to *trans* effects, further reducing the power to detect *cis*-acting signals at individual target genes[4].

However, interferon response involves rapid upregulation of a broad transcriptional regulatory network of genes with diverse biological functions, only a subset of which might have a direct causal effect on SLE. This is supported by the fact that among the 38 interferon response genes (category II), only *OAS1, ADAR, PSMB8, SAMHD1* and the IRF transcription factors have been implicated in causing primary immunodeficiencies (Fig. 3B). The remaining interferon response genes might thus be better thought of as biomarkers of the complex effect of interferon signalling on multiple parts of the immune system[50,69]. Similarly, it has been previously shown that variants in the *IL6R* region that are associated with circulating C-reactive protein (CRP) concentrations, are also associated with coronary artery disease (CAD) risk[70], but variants in the *CRP* region are not[71]. Thus, plasma levels of CRP do not seem to have a direct causal effect on CAD risk, but can still act as a molecular readout (biomarker) of the *IL6R*-mediated inflammatory response that does seem to have a causal effect[72]. These observations suggest that widespread horizontal pleiotropy in gene regulatory networks could be a general property of *trans*-QTLs and could help explain why using *trans*-pQTL signals in Mendelian randomisation analysis has had low specificity for identifying known drug targets[73,74]. Instead, we propose that target genes identified from large-scale *trans*-QTL studies could be better thought of as drug response biomarkers for drugs targeting the *cis* gene responsible for the *trans* association[8].

A limitation of our *trans*-eQTL analysis is its susceptibility to cross-mappability artefacts (Supplementary Data 6). While heuristic approaches have been developed to filter such artefacts *post hoc*, these approaches are not guaranteed to remove all cross-mappability effects and might be too conservative at other loci[29]. Cross-mappability artefacts also tend to replicate well in independent cohorts[29]. Furthermore, as the sample size of *trans*-eQTL studies increases, the power to detect subtle cross-mappability effects as putative *trans*-eQTLs also increases. To avoid these false positives, we used a very conservative strategy of requiring each *trans*-eQTL locus to have at least five independent target genes that all pass the cross-mappability filter. As a result, we likely missed many true *trans*-eQTLs regulating single or few target genes (e.g., *trans*-eQTL effect near the CIITA transcription factor on multiple HLA genes that has been replicated in several independent studies[6,31,75–77], Supplementary Data 1, Supplementary Fig. 8). Future large-scale *trans*-eQTL studies will likely require the development of novel methods to properly adjust for cross-mappability, such as explicit modelling of transcript compatibility read counts between *cis* and trans target genes[78].

While large-scale *trans*-eQTL studies using both bulk and single-cell measurements are likely to continue for easily accessible tissues such as whole blood (e.g., eQTLGen Phase 2[79]), it seems unlikely that we will be able to perform *trans*-eQTL studies comprising tens of thousands of individuals for all disease-relevant cell types and contexts. A promising alternative is to use arrayed CRISPR screens or single-cell approaches to identify downstream gene-regulatory effects of disease-associated genes or individual genetic variants[46,80,81].

## Methods

### Datasets, samples and ethics

We used genotype and gene expression data from ALSPAC[31,82,83], TwinsUK[84], CoLaus[85,86], GEUVADIS[87], MRCA[88], MRCE[88], GENCORD[89], GTEx v8[17] and CAP[90] studies. For replication, we used data from the MAGE cohort[30]. The RNA sequencing and genotype data from the GEUVADIS and MAGE studies was publicly available as part of the 1000 Genomes project. For the other studies, we applied for access to individual-level data via relevant data access committees (DACs), explaining the aim of our project and the intent to publicly share meta-analysis summary statistics. Informed consent was obtained when research participants joined the ten studies listed above. The use of the CAP data for this project was approved by the National Heart, Lung and Blood Institute DAC. The use of the GTEx data for this project was approved by the National Human Genome Research Institute DAC. The use of the GENCORD data for this project was approved by the GEN-CORD DAC. The use of the MRCA and MRCE data for this project was approved by the Gabriel Consortium DAC. The use of TwinsUK data for this project was approved by the TwinsUK Resource Executive Committee. The use of the ALSPAC data for this project was approved by the ALSPAC Executive Committee. For the ALSPAC cohort, ethical approval for the study was obtained from the ALSPAC Ethics and Law Committee and the Local Research Ethics Committees. Consent for biological samples has been collected in accordance with the Human Tissue Act (2004). The CoLaus study was approved by the Institutional Ethics Committee of the University of Lausanne. Single-cell RNA-seq samples were sourced ethically, and their research use was in accord with the terms of informed consent under an institutional review board/ethics committee-approved protocol (UK Regional Ethics Committee approval granted to work at Wellcome Sanger Institute, protocol reference number 15/NW/0282; project was approved by the Ethics on Research Committee of the Institute of Neurobiology at Universidad Nacional Autonoma de Mexico (UNAM), with the approval number 110.H.).

### Statistics and reproducibility

We performed genome-wide *trans*-eQTL meta-analysis across 3734 samples (2238 female, 1496 male) from 9 cohorts (Supplementary Table 1). Sex was assigned based on X and Y chromosome gene expression and genotype data as described previously[91]. We included all samples available to us into the discovery meta-analysis. We excluded a small number of samples due to issues with gene expression or genotype data quality (see below). To avoid potential confounding by population stratification, we exclude the 87 YRI ancestry individuals from the GEUVADIS dataset and 34 diverse ancestry samples from the GTEx dataset (defined based on genotype principal components). No statistical method was used to pre-determine sample size. However, evidence from several previous *trans*-eQTL studies with comparable or smaller sample sizes suggest that we should have sufficient power to detect *trans*-eQTLs with large effects[7,9,11–16,18,19]. Due to limited sample size of our study and significant computational cost involved, we did not perform sex-stratified *trans*-eQTL analysis. Biological sex was included as a covariate in association testing. We used power calculations to estimate if we had sufficient power to replicate the USP18 *trans*-eQTL effect in the MAGE cohort (Supplementary Note 2). The experiments were not randomized. The investigators were not blinded to allocation during experiments and outcome assessment.

### Genotype data quality control and imputation

**Pre-imputation quality control.** Genotype imputation was performed as described previously[40]. Briefly, we lifted coordinates of the genotyped variants to the GRCh38 build with CrossMap v0.4.1[92]. We aligned the strands of the genotyped variants to the 1000 Genomes 30x on GRCh38 reference panel[93] using Genotype Harmonizer[94]. We excluded genetic variants with Hardy–Weinberg $p < 10^{-6}$, missingness $> 0.05$ and minor allele frequency $<0.01$ from further analysis. We also excluded samples with more than 5% of their genotypes missing.

**Genotype imputation and quality control.** Most of the datasets were imputed using the 1000 Genomes reference panel based on the

GRCh38 genome version. CoLaus dataset was imputed using the TOPMed Imputation Server[95–97], while still aligning with the same reference genome version. Additionally, GEUVADIS, GTEx and MAGE cohorts utilised whole genome sequencing data aligned to the GRCh38 reference genome.

We pre-phased and imputed the microarray genotypes to the 1000 Genomes 30x on GRCh38 reference panel[93] using Eagle v2.4.1[98] and Minimac4[96]. We used bcftools v1.9.0 to exclude variants with minor allele frequency (MAF) < 0.01 and imputation quality score $R^2 < 0.4$ from downstream analysis. The genotype imputation and quality control steps are implemented in eQTL-Catalogue/genimpute (v22.01.1) workflow available from GitHub. Subsequently, we used QCTOOL v2.2.0 to convert imputed genotypes from VCF format to bgen format for trans-eQTL analysis with regenie.

### Gene expression data

**Studies.** We used gene expression data from seven RNA-seq studies (TwinsUK[84], CoLaus[85,86], GEUVADIS[87], GENCORD[89], GTEx v8[17], CAP[90], MAGE[30]) and three microarray studies (ALSPAC[31,82,83], MRCA[88] and MRCE[88]).

**RNA-seq quantification and normalisation.** RNA-seq data were pre-processed as described previously[91]. Briefly, quantification of the RNA-seq data was performed using the eQTL-Catalogue/rnaseq workflow (v22.05.1) implemented in Nextflow. Before quantification, we used Trim Galore v0.5.0 to remove sequencing adapters from the fastq files. For gene expression quantification, we used HISAT2[99] v2.2.1 to align reads to the GRCh38 reference genome (Homo_sapiens.GRCh38.dna.primary_assembly.fa file downloaded from Ensembl). We counted the number of reads overlapping the genes in the GENCODE V30 reference transcriptome annotations with featureCounts v1.6.4.

We excluded all samples that failed the quality control steps as described previously[91]. We normalised the gene counts using the conditional quantile normalisation (cqn) R package v1.30.0 with gene GC nucleotide content as a covariate. We downloaded the gene GC content estimates from Ensembl biomaRt and calculated the exon-level GC content using bedtools v2.19.0[100]. We also excluded lowly expressed genes, where 95 per cent of the samples within a dataset had transcripts per million (TPM)-normalised expression less than 1. Subsequently, we used the inverse normal transformation to standardise quantification estimates. Normalisation scripts together with containerised software are publicly available at https://github.com/eQTL-Catalogue/qcnorm.

**Microarray data processing.** Gene expression from 877 individuals in the ALSPAC cohort was profiled using Illumina Human HT-12 V3 BeadChips microarray. We used the normalised gene expression matrix from the original publication[31]. In the MRCA cohort, gene expression from 327 individuals was profiled using the Human Genome U133 Plus 2.0 microarray. We downloaded the raw CEL files from ArrayExpress (E-MTAB-1425) and normalised the data using the Robust Multi-Array Average (RMA) method from the affy Bioconductor package[101]. In the MRCE cohort, gene expression from 484 individuals was profiled using the Illumina Human-6 v1 Expression BeadChip. As raw data was unavailable, we downloaded the processed gene expression matrix from ArrayExpress (E-MTAB-1428). In all three microarray datasets, we applied inverse normal transformation to each probe before performing trans-eQTL analysis. If there were multiple probes mapping to the same gene, the probe with the highest average expression was used.

### Trans-eQTL mapping and meta-analysis

We performed independent quality control and normalisation on all datasets and only included 18,792 protein coding genes in the analysis.

Trans-eQTL analysis was conducted separately on each dataset with regenie[102]. For studies containing related samples (TwinsUK, MRCA and MRCE) and ALSPAC, both step 1 and step 2 commands were employed, while for other datasets with a smaller number of unrelated samples (Supplementary Table 1), regenie was run in the linear regression mode (step 2 only). We used sex and six principal components of the normalised gene expression matrix and six principal components of genotype data as covariates in the trans-eQTL analysis. All scripts used to run trans-eQTL are publicly available at https://github.com/freimannk/regenie_analysis. Subsequently, we performed an inverse-variance weighted meta-analysis across studies. Meta-analysis workflow is available at https://github.com/freimannk/regenie_metaanalyse.

We used a cis window of ± 5 Mb to assign identified eQTLs into cis and trans eQTLs. To determine significant loci, we excluded variants proximal (± 1.5 Mb) to the most highly associated variant per gene. This approach allowed us to identify distinct and robust signals while mitigating potential confounding effects from nearby variants. By applying these filters, we found 79 trans-eQTLs loci at a suggestive p-value threshold of $1 \times 10^{-11}$.

### Accounting for cross-mappability

A major source of false positives in trans-eQTL analysis is cross-mappability, whereby RNA-seq reads from gene A erroneously align to gene B, leading to very strong apparent trans-eQTL signals[11,29]. To exclude potential cross-mappability artefacts, we excluded all trans-eQTLs where there was high cross-mappability (cross-mappability score from Saha et al.[29] > 1) between the trans-eQTL target gene and at least one protein coding gene in the cis region (± 1.5 Mb) of the trans-eQTL lead variant. Since some of the strongest cross-mappability artefacts affected one or few target genes (Supplementary Data 6), we further restricted our analysis to trans-eQTL loci that had five or more target genes with $p < 5 \times 10^{-8}$ and cross-mappability score <1.

### Random-effect meta-analysis

To further assess the robustness of our meta-analysis results, we performed a random-effects meta-analysis on the ten lead variants identified by our primary analysis. We used the DerSimonian-Laird method implemented in PyMARE. We estimated the between-study variance ($\tau^2$) and assessed statistical significance using a z-score and a two-tailed p-value. All of the associations remained significant using the random-effect model (Supplementary Data 1).

### Replication of trans-eQTL associations

**MAGE.** Since we used somewhat arbitrary thresholds to define the initial set of 10 loci (lead $p < 1 \times 10^{-11}$, five or more targets with $p < 5 \times 10^{-8}$), we sought to replicate our findings in an independent Multi-ancestry Analysis of Gene Expression (MAGE)[30] cohort. MAGE consisted of data from 731 lymphoblastoid cell lines from the 1000 Genomes project, 682 of which also had whole genome sequencing data available. We used two strategies to assess replication. First, we assessed if the lead variant-gene pair was nominally significant ($p < 0.05$) in the replication dataset with concordant direction of effect. Based on this criterion, 7/10 loci replicated (Supplementary Data 1). Secondly, since all of our loci had multiple target genes, we used the pi1 statistic to estimate the proportion of FDR < 5% target gene at each locus that had a non-null p-value in the replication dataset[103]. We used the qvalue R package[104] to calculate pi1 = 1-qvalue(5% FDR trans gene p-values)$pi0. For 3/10 loci, the proportion of non-null p-values was > 0.5 (Supplementary Data 1). Note that replication in an independent cohort does not help to reduce false positives due to cross-mappability, as cross-mappability artefacts tend to be highly replicable[29].

**eQTLGen consortium.** The eQTLGen Consortium is an initiative to investigate the genetic architecture of blood gene expression and to

understand the genetic basis of complex traits. We used interim summary statistics from eQTLGen phase 2, wherein a genome-wide eQTL analysis has been performed in 52 cohorts, representing 43,301 individuals.

All 52 cohorts performed cohort-specific analyses as outlined in the eQTLGen analysis cookbook (https://eqtlgen.github.io/eqtlgen-web-site/eQTLGen-p2-cookbook.html). Genotype quality control was performed according to standard bioinformatics practices and included quality metric-based variant and sample filtering, removing related samples, ethnic outliers and population outliers. Genotype data was converted to genome build GRCh38 if not done so already and the autosomes were imputed using the 1000 Genomes 30x on GRCh38 reference panel[93] (all ancestries) using the eQTLGen imputation pipeline (eQTLGen/eQTLGenImpute).

Like the genotype data, gene expression data was processed using the eQTLGen data QC pipeline (eQTLGen/DataQC). For array-based datasets, we used the results from the empirical probe mapping approach from our previous study[6] to connect the most suitable probe to each gene which has previously been to show expression in the combined BIOS whole blood expression dataset. Raw expression data was further normalized in accordance with the expression platform used (quantile normalization for Illumina expression arrays and TMM[105] for RNA-seq) and inverse normal transformation was performed. Gene expression outlier samples were removed and gene summary information was collected for filtering at the central site. Samples for whom there were mismatches in genetically inferred sex, reported sex, or the expression of genes encoded from sex chromosomes, were removed. Similarly, samples with unclear sex, based on genetics or gene expression were removed.

An adaptation of the HASE framework[106] was used to perform genome-wide meta-analysis. For genome-wide eQTLs analysis, this limits the data transfer size while ensuring participant privacy. At each of the cohorts, the quality controlled and imputed data was processed and encoded so that the individual level data can no longer be extracted, but while still allowing effect sizes to be calculated for the linear relationship between variants and gene expression (eQTLGen/ConvertVcf2Hdf5 and eQTLGen/PerCohortDataPreparations).

Centrally, the meta-analysis pipeline was run on the 52 cohorts. The pipeline which performs per cohort calculations of effect sizes and standard errors and the inverse variance meta-analysis is available at eQTLGen/MetaAnalysis. We included 4 genetic principal components, 20 gene expression principal components and other technical covariates (e.g., RNA integrity number) where available. Per every dataset, genes were included if the fraction of unique expression values was equal or greater than 0.8, variants were included based on imputation quality, Hardy–Weinberg equilibrium (HWE) and minor allele frequency (MAF) (Mach R2 ≥ 0.4, HWE $p \geq 1 \times 10^{-6}$ and MAF ≥ 0.01). In an additional step, genes were filtered to include only those genes that were available in at least 50% of the cohorts and 50% of the samples.

**OneK1K**. The OneK1K dataset consisted of single-cell RNA sequencing (scRNA-seq) data from naïve B cell samples collected from 844 donors. We relied on the original cell type annotations provided by the authors. Following the same approach as in the primary analysis, the data were processed using regenie in linear regression mode (step 2 only). For the *trans*-eQTL analysis, we included sex, six principal components of the normalized, rank-based inverse normal-transformed gene expression matrix, and six principal components of genotype data as covariates.

### Differential gene expression in SLE cases *versus* controls
We re-analysed the microarray gene expression data from Banchereau et al.[49] to explore differential gene expression between SLE cases and controls. After downloading the processed data from GEO (GSE65391), we selected one sample from each individual for our analysis based on

their earliest recorded visits. The filtered dataset comprised a total of 204 samples, including 46 samples from healthy individuals and 158 samples from individuals diagnosed with SLE. We also applied the inverse normal transformation to standardise the gene expression values. Subsequently, we used the Python statsmodels[107] module to fit a linear model to identify genes that were differentially expressed between SLE cases and controls. We included gender, age and batch as covariates in all models.

### Overlap between *USP18* target genes and GWAS hits for SLE
We downloaded the list of prioritised target genes for the Yin et al. GWAS study (GCST011956) from the Open Targets Genetics Portal. We combined the list of genes prioritised by either the L2G or the closest gene approach, yielding $n = 109$ target genes. We then overlapped these target lists with the list of 50 *trans*-eQTL targets for the *USP18* locus (FDR < 5%).

### Single-cell differential gene expression in resting and stimulated B-cells
**Sample collection, cell isolation and cryopreservation.** Blood samples were collected from five healthy Mexican individuals (three males and two females). Peripheral blood mononuclear cells (PBMCs) were isolated using Vacutainer CPT tubes, according to manufacturer instructions. Samples were cryopreserved in RPMI 1640 culture media (Sigma), Fetal Bovine Serum (FBS) and Dimethyl sulfoxide (DMSO) and stored at −80 °C for 24 h, before being transferred to liquid nitrogen.

**Thawing and stimulation.** Cryopreserved PBMCs were thawed quickly and washed in 14 mL of room temperature complete RPMI 1640 media (10% FBS, 1% Penicillin-Streptomycin, 1% L-Glutamine). Cells were incubated at 37 °C, 5% $CO_2$ for 2 h. Cells were then stimulated with interferon alpha (IFN-α, Bio-techne) and R-848 (Resiquimod, Cambridge Bioscience) at a working concentration of 1000 U/mL and 2 μg/ml, respectively. Cells were incubated at 37 °C, 5% CO2 and harvested after 16 h, 40 h and 64 h of stimulation. Unstimulated cells were kept in culture without any stimuli for 16 h (i.e., 0 h of activation).

**Multiplexing, CITE-seq staining & scRNA-seq.** Upon harvesting, cells were resuspended in a cell staining buffer (Biolegend) and cell hashing and genotype-based multiplexing was performed. Donors of the same stimulation condition were mixed at equal ratios (each pool corresponded to a mix of cells from four to five different individuals). These pools were stained with the TotalSeq-C Human Universal Cocktail, V1.0 (137 cell surface proteins (CSP), Biolegend), in addition to a unique hashtag antibody oligonucleotide (HTO, Biolegend) which corresponds to the stimulation condition pool. After staining and washing, all stimuli condition pools were pooled together at equal ratios. This pool was then stained with live/dead dye 4,6-diamidino-2-phenylindole (DAPI, Biolegend) and dead cells were removed using fluorescence-activated cell sorting.

Cells were next processed using the 10X Genomics Immune Profiling 5′ high-throughput (HT) v2 kit, as specified by the manufacturer's instructions. $1.15 \times 10^5$ cells were loaded into each inlet of a 10X Chromium X to create Gel Bead-in-emulsions (GEMs). Two 10X HT reactions were loaded per time point of sample processing (targeted recovery was 40,000 cells per 10X reaction). Reverse transcription was performed on the emulsion, after which cDNA and CITEseq supernatant were purified, amplified and used to construct RNA-sequencing and CSP sequencing libraries, respectively. These RNA and CSP libraries were sequenced at a 5:1 ratio, respectively, using the Illumina NoveSeq 6000 S4, with 100-bp paired-end reads and all 10X reactions were mixed at equal ratios and sequenced across two lanes.

**Deconvolution of single cells by genotype.** Each 10X reaction comprised a mix of cells from unrelated individuals. Thus, natural genetic

variation was used to assign cells to their respective individuals. First, a list of common exonic variants was compiled from the 1000 Genomes Project phase 3 exome-sequencing data (MAF > 0.05). Next, cellSNP (v1.2.1) was used to generate pileups at the genomic location of these variants. These pileups, in combination with the variants called from genotyping in each individual, were used as an input for Vireo[108] (v0.5.7). If any cell had less than 0.9 posterior probability of belonging to any individual or were of mixed genotypes they were labelled as 'unassigned' and 'doublets', respectively, and removed from downstream analysis.

**Data processing and quality controls.** Raw scRNA-seq and CITE-seq data were processed using the Cell Ranger Multi pipeline (v7.0.0, 10x Genomics). In brief, RNA and CSP library reads were first assigned to cells. RNA reads were then aligned to the GRCh38 human reference genome and CSP antibody reads were matched to the provided list of known barcodes. Ensembl version 93 was used as a reference for gene annotation, and gene expression was quantified using reads assigned to cells and confidently mapped to the genome. Additionally, Cell Ranger multi was used to deconvolute samples based on HTOs. It uses an algorithm which employs a latent variable model over a state space composed of each HTO used in the experiment to assign each cell to a stimulation condition or as a doublet.

Results from RNA and CSP quantification in Cell Ranger were imported into RStudio (v4.3.1) and analysed using Seurat (v5.0.1). Any cell identified as doublet or unassigned by Vireo and or antibody hashtag deconvolution method were excluded. 10X reactions were split by time point and stimuli condition. Cells with 1.5–2.5 median absolute deviations below the median of genes and counts detected were discarded. Additionally, cells with 3–4 median absolute deviation above the median for the percentage of mitochondrial reads detected were discarded. The resulting cells were then annotated by Azimuth[109] (v0.5.0), using the Azimuth PBMC reference dataset that was generated as part of the Hao and Hao et al., 2021 paper[109].

**Pseudobulk and normalisation.** Raw counts were pseudobulked by Azimuth annotated level 1 cell types (CD4, CD8, B, Mono, DC, NK, Other and Other T) per donor, per time point and per stimulation condition, via edgeR[110] (v4.0.16). Pseudobulked raw counts were then counts per million (CPM) normalised and $\log_2$ transformed with edgeR.

**Reporting summary**
Further information on research design is available in the Nature Portfolio Reporting Summary linked to this article.

## Data availability
The whole genome sequencing data for the GEUVADIS and MAGE studies was downloaded from the 1000 Genomes website (https://www.internationalgenome.org/data-portal/data-collection/30x-grch38). The GEUVADIS RNA-seq data was downloaded from the European Nucleotide Archive (ENA) under accession PRJEB3366. The MAGE RNA-seq data was downloaded from the ENA (accession PRJNA851328). The genotype and RNA-seq data from the GENCORD study was downloaded from European Genotype-phenotype Archive (EGA) under accessions EGAD00001000425 and EGAD00001000428. The microarray gene expression data from the MRCA and MRCE studies was downloaded from ArrayExpress (accessions E-MTAB-1425 and E-MTAB-1428) and the genotype data was downloaded from EGA (accession EGAS00000000137). The gene expression and genotype data from GTEx and CAP studies was downloaded from dbGaP (accessions phs000424.v8.p2 [https://www.ncbi.nlm.nih.gov/projects/gap/cgi-bin/study.cgi?study_id=phs000424.v8.p2] and phs000481.v3.p2 [https://www.ncbi.nlm.nih.gov/projects/gap/cgi-bin/study.cgi?study_id=phs000481.v3.p2]). The RNA-seq data from

the TwinsUK study was downloaded from EGA (accession EGAD00001001086) and genotype data was obtained from TwinsUK (https://twinsuk.ac.uk/researchers/access-data-and-samples/request-access/). The gene expression data from SLE cases and controls is available from GEO under accession code GSE65391. The informed consent obtained from ALSPAC participants does not allow the microarray and genotype data to be made freely available through any third party maintained public repository. However, data used for this study can be made available on request to the ALSPAC Executive. The ALSPAC data management plan describes in detail the policy regarding data sharing, which is through a system of managed open access. Full instructions for applying for data access can be found here: http://www.bristol.ac.uk/alspac/researchers/access/. The ALSPAC study website contains details of all the data that are available (http://www.bristol.ac.uk/alspac/researchers/our-data/). The RNA-seq and genotype data from the CoLaus cohort can be accessed by directly contacting the cohort (https://www.colaus-psycolaus.ch/professionals/how-to-collaborate/). The MetaLCL full *trans*-eQTL meta-analysis summary statistics are available from the eQTL Catalogue FTP server (https://www.ebi.ac.uk/eqtl/Data_access/) and additional documentation is available on the project website (https://github.com/AlasooLab/MetaLCL). Source data are provided with this paper.

## Code availability
The MetaLCL *trans*-eQTL analysis workflows are available from https://github.com/freimannk/regenie_analysis, the MetaLCL meta-analysis workflow is available from https://github.com/freimannk/regenie_metaanalyse. Additional documentation and code used to generate figures in the paper is available from https://github.com/AlasooLab/MetaLCL. The eQTL Catalogue genotype imputation, RNA-seq processing and data normalisation workflows are available from https://github.com/eQTL-Catalogue/genimpute, https://github.com/eQTL-Catalogue/rnaseq and https://github.com/eQTL-Catalogue/qcnorm. The eQTLGen genotype imputation, data quality control, data preparation and meta-analysis workflows are available from https://github.com/eQTLGen/eQTLGenImpute, https://github.com/eQTLGen/DataQC, https://github.com/eQTLGen/ConvertVcf2Hdf5, https://github.com/eQTLGen/PerCohortDataPreparations and https://github.com/eQTLGen/MetaAnalysis.

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

## Acknowledgements

Most of the analysis presented in the paper were performed at the High Performance Computing Center, University of Tartu. We are extremely grateful to all the families who took part in the ALSPAC study, the mid-wives for their help in recruiting them, and the whole ALSPAC team, which includes interviewers, computer and laboratory technicians, cle-rical workers, research scientists, volunteers, managers, receptionists and nurses. This work received support from Christian Molina-Aguilar, Carina Uribe Díaz, and Alejandra Castillo Carbajal. We thank the FACS facility and sequencing pipelines at Wellcome Sanger Institute for their assistance in data generation, and the Human Genetics Informatics for their support in sequence data processing. We thank Bess L. Chau for assistance in the experiments involving generation of single-cell RNA-seq data from stimulated PBMCs. We thank Karatug Ozan Bircan, Abayomi Mosaku, Laura Harris and Helen Parkinson for assistance with hosting the full *trans*-eQTL summary statistics on the eQTL Catalogue FTP server. We thank Peep Kolberg for providing access to processed data from the OneK1K cohort. The cell membrane icon on Figs. 2C and 3A by Servier (https://smart.servier.com/) is licensed under Creative Commons Attribution 4.0. K.F. and L.S. were supported by a grant from Open Targets (grant no. OTAR2069). K.A. was supported by the Estonian Research Council (grant nos. PSG415 and MOB3ERC115). K.A. also received funding from the European Union's Horizon 2020 research and innovation program (grant no. 825775). G.T., C.P.J., and T.S.R. were supported by Open Targets (grant no. OTAR2064) and the Wellcome Trust (ref. 220540/Z/20/A, 'Wellcome Sanger Institute Quinquennial Review 2021-2026). S.B. was supported by the Swiss National Science Foundation (grant no. 310030_152724/1). K.K. was supported by the Estonian Research Council (grant no. PRG1117). A.M.-R. was supported by CONACYT-FORDECYT-PRONACES (grant no. 11311) and Programa de Apoyo a Proyectos de Investigación e Innovación Tecnológica–Universidad Nacional Autónoma de México (PAPIIT-UNAM) IN218023. A.L.H.-L is a doctoral student from Programa de Doctorado en Ciencias Biomédicas, Universidad Nacional Autónoma de México (UNAM), and she receives fellowship 790972 from Consejo Nacional de Humanidades, Ciencias y Tecnologías CONAHCYT, México. The UK Medical Research Council and Wellcome (Grant no. 217065/Z/19/Z) and the University of Bristol provide core support for ALSPAC. ALSPAC GWAS data was generated by Sample Logistics and Genotyping Facil-ities at Wellcome Sanger Institute and LabCorp (Laboratory Corporation of America) using support from 23andMe. This publication is the work of the authors and they will serve as guarantors for the contents of this paper.

## Author contributions

K.F. performed *trans*-eQTL analysis on nine cohorts and also performed the meta-analysis across cohorts. A.B. performed *trans*-eQTL analysis on the CoLaus cohort. K.F., J.C., E.R.H., J.C.M., N.N., H.O., L.S., M.C.T., G.T., K.K., and K.A. interpreted the results and prioritised follow-up analyses. T.S.R. generated and analysed single-cell RNA-seq data under C.P.J. and G.T. supervision. A.L.H.-L. collected blood samples for single-cell RNA-seq under A.M.-R. supervision. R.W. performed replications in the eQTLGen Consortium under U.V. and L.F. supervision. S.B. and K.A. supervised the research. K.F. and K.A. wrote the manuscript with feed-back from all authors.

## Competing interests

J.C. is an employee of Pfizer. E.R.H., M.C.T., and J.C.M. are employees of Bristol Myers Squibb. H.O. is an employee of GSK. N.N. was an employee of GSK while this work was conducted. The other authors declare no competing interests.

## Additional information

# eQTLGen Consortium

**eQTLGen** Habibul Ahsan[16], Philip Awadalla[17], Alexis Battle[18], Frank Beutner[19], Cornelis Blauwendraat[20,21], Collins Boahen[22,23], Toni Boltz[24], Marc Jan Bonder[5,6], Dorret I. Boomsma[25,26,27], John Budde[28,29], Katie L. Burnham[7], John Chambers[30,31,32], Evans Cheruiyot[33], Surya B. Chhetri[34], Annique Claringbould[5,35], Carlos Cruchaga[28,29,36,37,38,39], Kensuke Daida[20,21,40], Emma E. Davenport[7], Patrick Deelen[5,6], Devin Dikec[28,29], Diptavo Dutta[41], Tõnu Esko[13], Radi Farhad[42], Aiman Farzeen[40,41,42,43,44,45], Marie-Julie Favé[46], Luigi Ferrucci[47], Lude Franke ®[5,6], Timothy M. Frayling[48], Koichi Fukunaga[49], J. Raphael Gibbs[50], Greg Gibson[51], Christian Gieger[44,45], Priyanka Gorijala[28,29], Marleen van Greevenbroek[52], Binisha Hamal Mishra[53,54,55], Takanori Hasegawa[56], Jouke Jan Hottenga[57,58], M. Arfan Ikram[59], Michael Inouye[60,61,62,63,64,65], Rick Jansen[66], Farzana Jasmine[67], Matt Johnson[28,29], Mika Kähönen[54,68], Muhammad Kibriya[67], Holger Kirsten[69,70], Julian C. Knight[71], Peter Kovacs[72,73], Knut Krohn[74], Viktorija Kukushkina[13], Vinod Kumar[22,23,75], Sandra Lapinska[76], Terho Lehtimäki[53,54,55], Yun Li[77,78,79], Markus Loeffler[70,80], Marie Loh[30,32,81,82], Leo-Pekka Lyytikäinen[53,54,55], Reedik Mägi[13], Javier Martin[83], Angel Martinez-Perez[84], Allan F. McRae[33], Joyce van Meurs[85,86], Lili Milani[13], Pashupati P. Mishra[53,54,55], Younes Mokrab[42], Grant W. Montgomery[33], Juha Mykkänen[87,88], Haroon Naeem[42], Sini Nagpal[51], Ho Namkoong[89], Matthias Nauck[90], Yukinori Okada[91,92,93,94,95], Roel Ophoff[24,77,96], Katja Pahkala[87,88,97], Bogdan Pasaniuc[24,76,98,99], Dirk S. Paul[60,61,100], Brenda W.J.H Penninx[66], Elodie Persyn[60,61,62], Annette Peters[45,101,102,103], Brandon Pierce[67], René Pool[26,58], Holger Prokisch[43,104,105], Laura Raffield[78], Venket Raghavan[70], Olli T. Raitakari[87,88,106,107], Emma Raitoharju[108,109], María Rivas-Torrubia[110], Ruth D. Rodríguez[110], Suvi P. Rovio[87,88], Jessie Sanford[28,29], Markus Scholz[70,80], Andrew Singleton[111], Eline Slagboom[112], José Manuel Soria[84,113], Juan Carlos Souto[84], Michael Stumvoll[103,114,115], Yun Ju Sung[28,29], Darwin Tay[30], Alexander Teumer[116,117,118], Joachim Thiery[80,119], Alex Tokolyi[7], Lin Tong[67], Anke Tönjes[114], Jan Veldink[120], Joost Verlouw[85], Peter M. Visscher[33], Uwe Völker[117,121], Urmo Võsa ®[13], Qingbo S. Wang[91,92,93], Robert Warmerdam ®[5,6], Stefan Weiss[117,121], Jia Wen[78], Harm-Jan Westra[5,6], Andrew R. Wood[122], Manke Xie[51], Dasha Zhernakova[5]

**DIRECT Brown** Andrew Brown[123], Théo Dupuis[123], Ana Viñuela[124]

**PRECISESADS Clinical Consortium** Marta E. Alarcón-Riquelme[110,125] & Guillermo Barturen[110,126,127]

[16]Biological Sciences Division, Institute for Population and Precision Health (IPPH), University of Chicago, Chicago, IL, USA. [17]Ontario Institute for Cancer Research, University of Toronto, Toronto, ON, Canada. [18]Department of Biomedical Engineering, Department of Computer Science, Department of Genetic Medicine, Johns Hopkins University, Baltimore, MD, USA. [19]Department of Internal Medicine/Cardiology, Heart Center Leipzig at Leipzig University, Leipzig, Germany. [20]Integrative Neurogenomics Unit, Laboratory of Neurogenetics, National Institute on Aging, National Institutes of Health, Bethesda, MD, USA. [21]Center for Alzheimer's and Related Dementias (CARD), National Institute on Aging and National Institute of Neurological Disorders and Stroke, National Institutes of Health, Bethesda, MD, USA. [22]Department of Internal Medicine and Radboud Institute of Molecular Life Sciences (RIMLS), Radboud University Medical Center, Nijmegen, the Netherlands. [23]Department of Internal Medicine and Radboud Center for Infectious Diseases (RCI), Radboud University Medical Center, Nijmegen, the Netherlands. [24]Department of Human Genetics, David Geffen School of Medicine, University of California Los Angeles, Los Angeles, CA, USA. [25]Department of Complex Trait Genetics, Center for Neurogenomics and Cognitive Research, Vrije Universiteit Amsterdam, Nijmegen, the Netherlands. [26]Amsterdam Public Health research institute, Amsterdam, the Netherlands. [27]Amsterdam Reproduction and Development (AR&D) Research Institute, Amsterdam, the Netherlands. [28]Department of Psychiatry, Washington University School of Medicine, St. Louis, MO, USA. [29]NeuroGenomics and Informatics Center, Washington University School of Medicine, St. Louis, MO, USA. [30]Lee Kong Chian School of Medicine, Nanyang Technological University, Singapore, Singapore. [31]Precision Health Research (PRECISE), Singapore, Singapore. [32]Department of Epidemiology and Biostatistics, School of Public Health, Imperial College London, London, UK. [33]Institute for Molecular Bioscience, The University of Queensland, Brisbane, QLD, Australia. [34]Department of Biomedical Engineering, Johns Hopkins University, Baltimore, MD, USA. [35]Department of Internal Medicine, Erasmus MC, Erasmus University Medical Center Rotterdam, Rotterdam, the Netherlands. [36]Department of Neurology, Washington University School of Medicine, St. Louis, MO, USA. [37]Knight Alzheimer Disease Research Center, Washington University School of Medicine, St. Louis, MO, USA. [38]Hope Center for Neurological Disorders, Washington University School of Medicine, St. Louis, MO, USA. [39]Dominantly Inherited Alzheimer Disease Network (DIAN), St. Louis, MO, USA. [40]Department of Neurology, Faculty of Medicine, Juntendo University, Tokyo, Japan. [41]Division of Cancer Epidemiology & Genetics, National Cancer Insititute, Bethesda, MD, USA. [42]Human Genetics Department, Sidra Medicine, Doha, Qatar. [43]Institute of Neurogenomics, Computational Health Center, Helmholtz Munich, Neuherberg, Germany. [44]Research Unit of Molecular Epidemiology, Helmholtz Zentrum München—German Research Center for Environmental Health, Neuherberg, Germany. [45]Institute of Epidemiology, Helmholtz Zentrum München—German Research Center for Environmental Health, Neuherberg, Germany. [46]Ontario Institute for Cancer Research, Toronto, ON, Canada. [47]Translational Gerontology Branch, National Institute on Aging, National Institutes of Health, Baltimore, MD, USA. [48]Department of Genetic Medicine and Development, CMU, University of Geneva, Geneva, Switzerland. [49]Division of Pulmonary Medicine, Department of Medicine, Keio University School of Medicine, Tokyo, Japan. [50]Computational Biology Group, Laboratory of Neurogenetics, National Institute on Aging, Bethesda, MD, USA. [51]Center for Integrative Genomics, Georgia Institute of Technology, Atlanta, GA, USA. [52]CARIM, Maastricht University, Maastricht, the Netherlands. [53]Department of Clinical Chemistry, Faculty of Medicine and Health Technology, Tampere University, Tampere, Finland. [54]Finnish Cardiovascular Research Center Tampere, Faculty of Medicine and Health Technology, Tampere University, Tampere, Finland. [55]Department of Clinical Chemistry, Fimlab Laboratories, Tampere, Finland. [56]M&D Data Science Center, Tokyo Medical and Dental University, Tokyo, Japan. [57]Neurological Disorder Research Center, Qatar Biomedical Research Institute (QBRI), Hamad Bin Khalifa University (HBKU), Qatar Foundation, Doha, Qatar. [58]Department of Biological Psychology, Vrije Universiteit Amsterdam, Amsterdam, the Netherlands. [59]Department of Epidemiology,

Erasmus MC University Medical Center, Rotterdam, the Netherlands. [60]British Heart Foundation Cardiovascular Epidemiology Unit, Department of Public Health and Primary Care, University of Cambridge, Cambridge, UK. [61]Victor Phillip Dahdaleh Heart and Lung Research Institute, University of Cambridge, Cambridge, UK. [62]Cambridge Baker Systems Genomics Initiative, Department of Public Health and Primary Care, University of Cambridge, Cambridge, UK. [63]Cambridge Baker Systems Genomics Initiative, Baker Heart and Diabetes Institute, Melbourne, VIC, Australia. [64]Health Data Research UK Cambridge, Wellcome Genome Campus and University of Cambridge, Cambridge, UK. [65]British Heart Foundation Centre of Research Excellence, University of Cambridge, Cambridge, UK. [66]Department of Psychiatry, Amsterdam Public Health (Mental Health program) and Amsterdam Neuroscience (Mood, Anxiety, Psychosis, Stress and Sleep Program) Research Institutes, Amsterdam UMC Location Vrije University Amsterdam, Amsterdam, the Netherlands. [67]Biological Sciences Division, Public Health Sciences, University of Chicago, Chicago, IL, USA. [68]Department of Clinical Physiology, Tampere University Hospital, Tampere, Finland. [69]LIFE—Leipzig Research Center for Civilization Diseases, Leipzig University, Leipzig, Germany. [70]Institute for Medical Informatics, Statistics and Epidemiology, Leipzig University, Leipzig, Germany. [71]Centre for Human Genetics, University of Oxford, Oxford, UK. [72]Integrated Research and Treatment Center (IFB) Adiposity Diseases, University of Leipzig, Leipzig, Germany. [73]Department of Medicine, University of Leipzig, Leipzig, Germany. [74]Medical Faculty, University of Leipzig, Leipzig, Germany. [75]Nitte (Deemed to Be University), Medical Sciences Complex, Nitte University Centre for Science Education and Research (NUCSER), Deralakatte, Mangalore, India. [76]Bioinformatics Interdepartmental Program, University of California Los Angeles, Los Angeles, CA, USA. [77]Department of Biostatistics, University of North Carolina at Chapel Hill, Chapel Hill, NC, USA. [78]Department of Genetics, University of North Carolina, Chapel Hill, Chapel Hill, NC, USA. [79]Department of Computer Science, University of North Carolina at Chapel Hill, Chapel Hill, NC, USA. [80]Leipzig Research Centre for Civilization Diseases, Leipzig University, Leipzig, Germany. [81]National Skin Centre, Research Division, Singapore, Singapore. [82]Genome Institute of Singapore, Agency for Science, Technology and Research, Singapore, Singapore. [83]Instituto de Parasitología y Biomedicina López-Neyra, Consejo Superior de Investigaciones Científicas (IPBLN-CSIC), Granada, Spain. [84]Unit of Genomics of Complex Diseases, Institut de Recerca Sant Pau (IR Sant Pau), Barcelona, Spain. [85]Department of Internal Medicine, Erasmus MC University Medical Center, Rotterdam, the Netherlands. [86]Department of Orthopaedics and Sportsmedicine, Erasmus MC University Medical Center, Rotterdam, the Netherlands. [87]Centre for Population Health Research, University of Turku and Turku University Hospital, Turku, Finland. [88]Research Centre of Applied and Preventive Cardiovascular Medicine, University of Turku, Turku, Finland. [89]Department of Infectious Diseases, Keio University School of Medicine, Tokyo, Japan. [90]Institute of Clinical Chemistry and Laboratory Medicine, University Medicine Greifswald, Greifswald, Germany. [91]Department of Genome Informatics, Graduate School of Medicine, the University of Tokyo, Tokyo, Japan. [92]Department of Statistical Genetics, Osaka University Graduate School of Medicine, Suita, Japan. [93]Laboratory for Systems Genetics, RIKEN Center for Integrative Medical Sciences, Yokohama, Japan. [94]Laboratory of Statistical Immunology, Immunology Frontier Research Center (WPI-IFReC), Osaka University, Suita, Japan. [95]Premium Research Institute for Human Metaverse Medicine (WPI-PRIMe), Osaka University, Suita, Japan. [96]Center for Neurobehavioral Genetics, Semel Institute for Neuroscience and Human Behavior, David Geffen School of Medicine, University of California Los Angeles, Los Angeles, CA, USA. [97]Paavo Nurmi Centre, Unit of Health and Physical Activity, University of Turku, Turku, Finland. [98]Department of Computational Medicine, David Geffen School of Medicine, University of California Los Angeles, Los Angeles, CA, USA. [99]Department of Pathology and Laboratory Medicine, David Geffen School of Medicine, University of California Los Angeles, Los Angeles, CA, USA. [100]Centre for Genomics Research, Discovery Sciences, BioPharmaceuticals R&D, AstraZeneca, Cambridge, UK. [101]Chair of Epidemiology, IBE, Faculty of Medicine, LMU Munich, Munich, Germany. [102]German Centre for Cardiovascular Research (DZHK), Partner Site Munich Heart Alliance, Munich, Germany. [103]German Center for Diabetes Research (DZD), Neuherberg, Germany. [104]School of Medicine, Institute of Human Genetics, Technical University of Munich, Munich, Germany. [105]German Center for Child and Adolescent Health (DZKJ), Partner Site Munich, Munich, Germany. [106]Department of Clinical Physiology and Nuclear Medicine, Turku University Hospital, Turku, Finland. [107]InFLAMES Research Flagship, University of Turku, Turku, Finland. [108]Molecular Epidemiology, Faculty of Medicine and Health Technology, Tampere University, Tampere, Finland. [109]Tampere University Hospital, Tampere, Finland. [110]Pfizer–University of Granada–Junta de Andalucía Centre for Genomics and Oncological Research, Granada, Spain. [111]Laboratory of Neurogenetics, National Institute on Aging, National Institutes of Health, Bethesda, MD, USA. [112]Section of Molecular Epidemiology, Department of Biomedical Data Sciences, Leiden University Medical Center, Leiden, the Netherlands. [113]Centre for Biomedical Network Research on Rare Diseases (CIBERER), Instituto de Salud Carlos III, Madrid, Spain. [114]Medical Department III-Endocrinology, Nephrology, Rheumatology, University of Leipzig Medical Center, Leipzig, Germany. [115]Helmholtz Institute for Metabolic, Obesity and Vascular Research (HI-MAG), Helmholtz Zentrum München, University of Leipzig and University Hospital Leipzig, Leipzig, Germany. [116]Department of Psychiatry and Psychotherapy, University Medicine Greifswald, Greifswald, Germany. [117]DZHK (German Center for Cardiovascular Research), Partner Site Greifswald, Greifswald, Germany. [118]Department of Population Medicine and Lifestyle Diseases Prevention, Medical University of Bialystok, Bialystok, Poland. [119]Institute of Laboratory Medicine, Clinical Chemistry and Molecular Diagnostics, University of Leipzig Medical Center, Leipzig, Germany. [120]Department of Neurology, UMC Utrecht Brain Center Rudolf Magnus, Utrecht, the Netherlands. [121]Interfaculty Institute of Genetics and Functional Genomics, University Medicine Greifswald, Greifswald, Germany. [122]College of Medicine and Health, University of Exeter, Exeter, UK. [123]Population Health and Genomics, University of Dundee, Dundee, Scotland, UK. [124]Biosciences Institute, Faculty of Medical Sciences, University of Newcastle, Newcastle upon Tyne, UK. [125]Institute of Environmental Medicine, Karolinska Institute, Stockholm, Sweden. [126]Department of Genetics, Faculty of Science, University of Granada, Granada, Spain. [127]Bioinformatics Laboratory, Biotechnology Institute, Centro de Investigación Biomédica, PTS, Centro de Investigación Biomédica, PTS, Granada, Spain.

