## [Transparent Peer Review file · Nature Communications]

Trans-eQTL mapping prioritises USP18 as a negative regulator of interferon response at a lupus risk locus

Corresponding Author: Dr Kaur Alasoo

Version 0:

Reviewer comments:

Reviewer #1

(Remarks to the Author)

Freimann and others investigate genetic regulation of gene expression using trans-eQTL mapping in LCLs. By analyzing 3,734 samples, the researchers identify four robust trans-eQTL loci, with a particular focus on the loci near USP18 gene. The study demonstrates that this loci has trans-eQTL effects that are context-specific, emphasizing the importance of cell-type-specific analyses in understanding genetic regulation. Through detailed descriptive data curation and analysis, the authors highlight the opportunities of interpreting trans-eQTL signals, especially in linking genetic variants to mechanisms for SLE.

1. The authors need to explain in greater detail why they chose to proceed with the locus near USP18? Currently the authors provide 1 linear description of 2 of 3 other loci. I would recommend that the authors emphasize that they "chose" USP18 locus for analysis. Currently the set up seems to be an agnostic QTL scan and follow up of the loci which would warrant the authors to investigate each of the loci (including HNF4G which I didn't find any major mention of). But the authors chose to proceed with USP18, which is absolutely fine but needs to be justified and mentioned like that.

2. The authors did a commendable job of attempting to identify the causal variant at the USP18 locus. But there is no cis-mediation, and the authors have only provided descriptive analysis of the variants. Without any functional or stronger in silico evidence it is hard to accept that this is the causal variant. In fact, could the causal variant be something that has not been imputed/typed like a structural variant? If the authors want to make this point about the causal variant and gene at this locus, I would strongly recommend a functional assay.

3. The overexpression of USP18 in only LCL in GTEx is curious. Is there a possibility of cross-mapping artefact with EBV genes?

4. As such the authors reinforce known results about the involvement of interferon in SLE. Even if there's a novel finding, the authors need to highlight it much strongly throughout the manuscript.

Minor comments:

Fig S2: What is the trans-gene target in the figure? I presume each panel is a variant-gene pair that is being shown for consistency?

Lines 148-150: Can the authors refer to some papers to substantiate this claim that the cis-mediation needs to happen in only LCLs?

Sometimes gene names are inconsistently italicized. E.g. USP18 in line 152

(Remarks on code availability)

(Remarks to the Author)

Freimann et al. conducted a comprehensive trans-eQTL meta-analysis using LCL samples, identifying four robust trans-eQTLs. One notable finding is the association of the USP18 locus with systemic lupus erythematosus (SLE). The SLE protective allele (rs4819670-C), which is in high linkage disequilibrium with a missense variant in USP18, correlates with reduced expression of genes involved in the Type I interferon pathway. Overall, the manuscript is well-written, and the results are reliable, having been replicated across multiple independent cohorts. All four trans-eQTL signals are closely related to transcription factors, providing valuable insights into their function and the genetic regulatory mechanisms underlying GWAS findings. Below are some suggestions and comments to help improve the quality of the manuscript.

1. Gene expression in three cohorts was quantified using a microarray platform, while others used RNA-seq. To address potential biases introduced by these different quantification methods, one possible approach is to include the quantification method (RNA-seq or microarray) as a covariate when calculating eQTLs, ensuring that differences in platforms do not confound the results. This adjustment would enhance the robustness and comparability of the findings across cohorts.
2. For the colocalization and fine-mapping analyses, relying solely on a single variant could be problematic, even though the overall signals in the locus were imputed using POEMColoc. A more robust approach would be to utilize the full summary statistics from the GWAS catalog for SLE, particularly from the trans-ancestry meta-analysis (GCST011096) or the EAS population dataset (GCST90011866). While the lead variant may not be included in these datasets, leveraging imputed data across the entire region would likely yield more reliable results, providing a more comprehensive picture of the genetic associations and reducing the risk of biased conclusions.
3. Given the relatively high heterogeneity index (I^2) reported in Table S2, the use of a random effects model would be appropriate to further confirm the associations.
4. Detailed summary statistics, including beta, standard error (SE), p-value, direction of effect across different cohorts, and I^2 etc., for the trans-eQTL target genes of USP18 should be included in the main table. As these represent the major findings of this manuscript, providing this information will enhance transparency, allow for better interpretation of the results, and enable readers to assess the consistency and robustness of the associations across cohorts.
5. It is unclear whether publicly available ChIP-seq data for USP18 exists. If such data is accessible, further analysis could provide valuable insights into the regulatory mechanisms associated with USP18. Specifically, ChIP-seq data could help identify transcription factors binding to the locus and elucidate how these interactions contribute to the observed trans-eQTL effects and the regulation of genes involved in the Type I interferon pathway.
6. For Figure 1B, as mentioned in the legend, the upper panel is described as indicating the number of trans associations detected. However, the Y-axis of the upper scatter plot is labeled as the number of genes, which is somewhat confusing. It would be helpful to clarify the meaning of each dot in the upper plot—whether it represents individual genes associated with different trans-eQTLs or some other metric?
7. For Figure 2A, the current visualization does not follow the standard format of a locuszoom plot. To improve clarity and interpretability, the authors should label the chromosome and include the surrounding genes in the region. Additionally, incorporating LD information and recombination rates for the locus would provide important context for understanding the genetic architecture and the relationships of the variants. These enhancements will make the figure more informative and align with standard practices for presenting GWAS data.
8. For Figure 2B, the authors should clearly specify the reference allele used to calculate the effect sizes. This information is crucial for interpreting the direction and magnitude of the associations presented in the figure.
9. For Figure 2C, the labeling of "type 1 interferon" should be corrected to "IFN-1" rather than "INF-I" to ensure accuracy and consistency in nomenclature. Similar corrections are needed for "INF-I receptor". Clear and precise labeling will improve the professionalism and readability of the figure.
10. For Table S3, what's the meaning of "AC" in the 7th column?
11. In line 154, the authors state that the identified USP18 trans-eQTL variant rs4819670 is in high LD with the missense variant rs3180408 in the USP18 gene in EUR and EAS populations. However, it is unclear whether rs4819670_C is in high LD with rs3180408_C or with rs3180408_T. Based on publicly available data from LDlink, rs4819670_C is in high LD with rs3180408_C. The authors should clarify this relationship explicitly to avoid confusion and ensure the accuracy of the interpretation regarding the functional and regulatory implications of these variants.
12. In lines 185-186, the authors state, "39/50 genes were also more highly expressed in peripheral blood mononuclear cells from SLE cases compared to controls (Table S4), consistent with the established role of increased interferon signaling in SLE." However, Table S4 indicates that 40 genes are labeled as highly expressed, not 39. This discrepancy should be addressed and corrected to ensure consistency between the text and the supplementary materials.

(Remarks on code availability)

I have checked the summary statistics. The data can be easily downloaded and include meta-analysis results.

Reviewer #3

(Remarks to the Author)

The author performed an extensive trans-eQTL analysis utilizing LCL samples, which lead to the identification of four robust loci. A significant discovery within this study is the link between the USP18 locus and SLE. Specifically, the SLE risk allele of rs4819670 at the USP18 locus correlates with increased expression levels of Type I interferon genes. While the study addresses an important gap in understanding the genetic variants contribution to the abnormal activation of interferon pathway, several major issues need to be addressed to strengthen the validity and impact of the findings.

1. How did the authors adjust for potential biases caused by the different measurement platforms—RNA-seq in some datasets and microarray in others—used in their analysis of data from various sources?

2. In line 154, multiple SNPs are in linkage disequilibrium with rs4819670 across both European and East Asian populations. How do the authors ensure that the functional effect is not being driven by these other SNPs instead? Further, the authors should provide all the LD variants information.

3. To date, extensive research on trans-eQTLs has provided novel insights into the genetic underpinnings of disease onset. Nonetheless, a critical gap in this field is the shortage of functional validation. For variants located in coding regions, assessing their functionality is relatively straightforward. The study identifies rs3180408 as a potential causal variant influencing the type I IFN signaling based on statistical associations without experimental validation. To substantiate this claim, the authors should conduct functional assays, such as CRISPR-Cas9-mediated gene editing and reporter assays, to experimentally validate the impact of rs3180408 on gene expression and type I IFN signaling pathways in B cells.

4. In line 265, the author points out that the USP18 trans-eQTL signal is context-specificity. Compared to bulk datasets, scRNA-seq eQTL datasets are less impacted by cell composition and are therefore ideal for trans-eQTL replication, could the author perform this analysis to further support their findings?

Minor comment:

1. In Figure 2 and Figure 3, “IFN” is misspelled as “INF”, the authors should carefully check these typos.

2. In line 162, “rs4819670 missense variant” should be “rs3180408 missense variant”?

3. Regarding Figure 2B, the authors need to explicitly state the reference allele utilized for calculating the effect sizes.

(Remarks on code availability)

Reviewer #4

(Remarks to the Author)

(Remarks on code availability)

README files are clear with enough instructions. I also checked the provided summary statistics files, and they could be downloaded and read by software. The information in the summary statistics is sufficient.

Version 1:

Reviewer comments:

Reviewer #1

(Remarks to the Author)

I have no further comments.

(Remarks on code availability)

Reviewer #2

(Remarks to the Author)

I am satisfied with the changes provided, as the authors have adequately addressed my concerns.

(Remarks on code availability)

I did not execute the code myself, but the documentation provided is clear and well-structured.

Reviewer #3

(Remarks to the Author)

The authors have provided thorough and satisfactory responses to my initial concerns, and I agree that the manuscript is suitable for publication in Nature Communications.

One minor suggestion for improvement is to enhance the resolution of the figures, especially those in the supplementary materials (FigS6C, FigS10...), to ensure clarity and readability.

(Remarks on code availability)

Reviewer #4

(Remarks to the Author)

(Remarks on code availability)

The README file is clear with enough instructions. We have checked the provided summary statistics file. They can be easily downloaded and read by software. The information in the summary statistics is also sufficient.

Reviewer #1 (Remarks to the Author):

Freimann and others investigate genetic regulation of gene expression using trans-eQTL mapping in LCLs. By analyzing 3,734 samples, the researchers identify four robust trans-eQTL loci, with a particular focus on the loci near USP18 gene. The study demonstrates that this loci has trans-eQTL effects that are context-specific, emphasizing the importance of cell-type-specific analyses in understanding genetic regulation. Through detailed descriptive data curation and analysis, the authors highlight the opportunities of interpreting trans-eQTL signals, especially in linking genetic variants to mechanisms for SLE.

Thank you.

1. The authors need to explain in greater detail why they chose to proceed with the locus near USP18? Currently the authors provide 1 linear description of 2 of 3 other loci. I would recommend that the authors emphasize that they “chose” USP18 locus for analysis. Currently the set up seems to be an agnostic QTL scan and follow up of the loci which would warrant the authors to investigate each of the loci (including HNF4G which I didn’t find any major mention of). But the authors chose to proceed with USP18, which is absolutely fine but needs to be justified and mentioned like that.

We have edited the sentence on page 5, lines 137-139 to clarify that we prioritised the *USP18* locus, because it was the only one that was in high LD ($r^2 > 0.9$) with an annotated GWAS hit in the Open Targets Genetics Portal.

“To prioritise the four trans-eQTL loci for follow-up analysis, we performed GWAS lookup using the Open Targets Genetics Portal^{β2} and found that only the USP18 locus lead variant (chr22_18166589_T_C, rs4819670) was in high LD ($r^2 > 0.9$) with an annotated GWAS hit.”

2. The authors did a commendable job of attempting to identify the causal variant at the USP18 locus. But there is no cis-mediation, and the authors have only provided descriptive analysis of the variants. Without any functional or stronger in silico evidence it is hard to accept that this is the causal variant. In fact, could the causal variant be something that has not been imputed/typed like a structural variant? If the authors want to make this point about the causal variant and gene at this locus, I would strongly recommend a functional assay.

We have now edited the main text (page 5-6, lines 174-184) to clarify that identifying the exact causal variant in this region remains challenging. In particular, we now also acknowledge that there could be additional untyped variants in the region that we are missing. We did consider a functional assay, but found that this would be extremely challenging due to the small expected effect size of the missense variant. The updated text now reads:

“However, identifying the exact causal variant remains challenging. While the absence of cis eQTL and splicing QTL evidence suggests that the likely causal mechanism is the rs3180408 missense variant, the missense variant was predicted to be benign by all tested variant effect

prediction tools available from Ensembl VEP³⁸. Alternatively, there could be other genetic variants in the region that are not captured by current genotype imputation reference panels (such as structural variants). One potential strategy to assess the functional impact of the rs3180408 missense variant on trans-eQTL target gene expression would be genome editing, but our power calculations (see Supplementary Text) suggest that this would be extremely challenging due to the small expected effect size of the variant.”

We have also removed the reference to the missense variant from the Abstract. The update text (lines 36-40) now reads:

“The trans-eQTL signal at the ubiquitin specific peptidase 18 (USP18) locus colocalised with a GWAS signal for systemic lupus erythematosus (SLE). USP18 is a known negative regulator of interferon signalling and the SLE risk allele increased the expression of 50 interferon-inducible genes, suggesting that the risk allele impairs USP18’s ability to effectively limit the interferon response.”

3. The overexpression of USP18 in only LCL in GTEx is curious. Is there a possibility of cross-mapping artefact with EBV genes?

We think cross-mappability with EBV genes is extremely unlikely. First, *USP18* is known to be strongly upregulated in response to interferon signalling and interferon signalling is highly active in LCLs. Secondly, BLAST identified only a very short overlap between the *USP18* gene and a non-coding junction region of the EBV genome (AH002364.2). Finally, visualising RNA-seq read coverage across the gene body of the *USP18* looked typical of most human protein coding genes and did not reveal evidence of cross-mappability (e.g. large number of reads aligning to only specific regions of the gene) (Response Figure 1).

Response Figure 1. RNA-seq read coverage across the USP18 gene in the GEUVADIS dataset. The read coverage has been stratified by the number of alternative alleles (0,1,2) of rs3180408-T a missense variant.

4. As such the authors reinforce known results about the involvement of interferon in SLE. Even if there's a novel finding, the authors need to highlight it much strongly throughout the manuscript.

We agree that the involvement of interferon response in SLE is firmly established. We believe the main novelty of our study is how it helps to resolve a GWAS hit at the *USP18* locus by demonstrating that it has a *trans*-regulatory effect on interferon response genes. This is interesting, because there are still only a handful of GWAS hits with demonstrated *trans*-eQTL activity. As a result, we were able to establish direction of effect at this locus - the allele that increased interferon signalling was also associated with increased risk of SLE. Secondly, we demonstrated how *trans*-eQTLs can influence the expression of many genes, only a small subset of which might mediate the causal effect on disease risk. As we highlight in the discussion, these observations suggest that widespread horizontal pleiotropy in gene regulatory networks could be a general property of *trans*-QTLs and could help explain why using *trans*-pQTL signals in Mendelian randomisation analysis has had low specificity for identifying known drug targets.

Minor comments:

Fig S2: What is the *trans*-gene target in the figure? I presume each panel is a variant-gene pair that is being shown for consistency?

We have now clarified Figures S2 by adding information about the locus name, lead variant and target gene for each of the four panels.

Figure S2. Forest plots showing cohort-specific effect size for the four *trans*-eQTL loci that replicated in the MAGE cohort. The points represent the *trans*-eQTL effect size estimates from regenie and the error bars represent 95% confidence intervals. The four panels correspond to the *BATF3*, *HNF4G*, *MYBL2* and *USP18* loci, their respective lead variants and lead target genes.

Lines 148-150: Can the authors refer to some papers to substantiate this claim that the cis-mediation needs to happen in only LCLs?

The LCL in this study were cultured in homogenous cell culture for many passages in the absence of other cell types from the blood samples from which they were originally derived. As a result, it is highly unlikely that these cell cultures still contain signalling molecules (e.g. proteins) produced by other cell types that could give rise to *trans*-eQTL effects.

On page 5, line 156, we have now add a reference to clarify this:

“Since the *trans*-eQTL effect was detected in monocultures of LCLs (Ryan et al. 2006),”.

Sometimes gene names are inconsistently italicized. E.g. *USP18* in line 152

This has now been fixed.

Reviewer #2 (Remarks to the Author):

Freimann et al. conducted a comprehensive trans-eQTL meta-analysis using LCL samples, identifying four robust trans-eQTLs. One notable finding is the association of the USP18 locus with systemic lupus erythematosus (SLE). The SLE protective allele (rs4819670-C), which is in high linkage disequilibrium with a missense variant in USP18, correlates with reduced expression of genes involved in the Type I interferon pathway. Overall, the manuscript is well-written, and the results are reliable, having been replicated across multiple independent cohorts. All four trans-eQTL signals are closely related to transcription factors, providing valuable insights into their function and the genetic regulatory mechanisms underlying GWAS findings. Below are some suggestions and comments to help improve the quality of the manuscript.

Thank you.

1. Gene expression in three cohorts was quantified using a microarray platform, while others used RNA-seq. To address potential biases introduced by these different quantification methods, one possible approach is to include the quantification method (RNA-seq or microarray) as a covariate when calculating eQTLs, ensuring that differences in platforms do not confound the results. This adjustment would enhance the robustness and comparability of the findings across cohorts.

We agree with the reviewer that combining gene expression data from different quantification platforms into a single regression analysis is likely to introduce biases. This is why we performed association testing separately in each cohort and subsequently meta-analysed the results. Notably, each cohort used only one quantification method (RNA-seq or microarray, see Table S1) so there was no need (nor was it even possible) to include the quantification method as a covariate in the linear model. We have now clarified this in the main text:

Page 3, lines 111-113: *“To avoid confounding by technical factors, we performed association testing separately in each cohort and meta-analysed the results (Figure 1A).”*

We agree that different quantification methods employed by different cohorts could introduce additional heterogeneity into our meta-analysis. However, we saw that all ten loci prioritised by our primary analysis remained significant when we used a more conservative random-effect meta-analysis that explicitly models heterogeneity between cohorts (see response to point 3 below).

2. For the colocalization and fine-mapping analyses, relying solely on a single variant could be problematic, even though the overall signals in the locus were imputed using POEMColoc. A more robust approach would be to utilize the full summary statistics from the GWAS catalog for SLE, particularly from the trans-ancestry meta-analysis (GCST011096) or the EAS population

dataset (GCST90011866). While the lead variant may not be included in these datasets, leveraging imputed data across the entire region would likely yield more reliable results, providing a more comprehensive picture of the genetic associations and reducing the risk of biased conclusions.

We downloaded the summary statistics for GCST011096 from the GWAS Catalog, and while we did detect colocalisation in that dataset as well (PP4 = 0.82), we were not confident in the colocalisation result due to the large number of missing variants and thus decided to not include it in the manuscript. However, we noticed that SLE was one of the phenotypes included in the recently released meta-analysis of 330 phenotypes across UK Biobank, FinnGen and Million Veterans Program (<https://mvp-ukbb.finnngen.fi/about>). Reassuringly, there was a robust association signal with SLE at *USP18* locus (p-value = 5.75×10^{-7}) and the most strongly associated variant was the rs3180408 *USP18* missense variant also prioritised by our analysis. There was also strong evidence of colocalisation between our *trans*-eQTL signal and the SLE GWAS signal from the FinnGen-UKBB-MVP meta-analysis (PP4 = 0.98).

We now report this in the main text (page 5; lines 142-145):

The colocalisation also replicated in an SLE GWAS meta-analysis across the UK Biobank (UKBB), FinnGen and Million Veterans Program (MVP) biobanks (Figure S3).

For your convenience, here is the newly added Figure S3:

Figure S3. Colocalisation between the SLE GWAS signal from the FinnGen + UKBB + MVP meta-analysis and the *USP18* *trans*-eQTL locus detected in our analysis.

3. Given the relatively high heterogeneity index (I^2) reported in Table S2, the use of a random effects model would be appropriate to further confirm the associations.

For the ten lead variant-gene pairs identified in our primary analysis, we have now also performed random-effect meta-analysis. All of the associations remained significant using the random-effect model. The results from the random-effect meta-analysis are now presented in the same Table S2.

We have also added the following paragraph to the Methods (page 15, lines 502-507):

Random-effect meta-analysis

To further assess the robustness of our meta-analysis results, we performed a random-effects meta-analysis on the ten lead variants identified by our primary analysis. We used the

DerSimonian-Laird method implemented in PyMARE. We estimated the between-study variance (τ^2) and assessed statistical significance using a Z-score and a two-tailed p-value. All of the associations remained significant using the random-effect model (Table S2).

4. Detailed summary statistics, including beta, standard error (SE), p-value, direction of effect across different cohorts, and I^2 etc., for the trans-eQTL target genes of USP18 should be included in the main table. As these represent the major findings of this manuscript, providing this information will enhance transparency, allow for better interpretation of the results, and enable readers to assess the consistency and robustness of the associations across cohorts.

We have now added the required information to Table S3.

5. It is unclear whether publicly available ChIP-seq data for USP18 exists. If such data is accessible, further analysis could provide valuable insights into the regulatory mechanisms associated with USP18. Specifically, ChIP-seq data could help identify transcription factors binding to the locus and elucidate how these interactions contribute to the observed trans-eQTL effects and the regulation of genes involved in the Type I interferon pathway.

USP18 is not known to be a transcription factor that binds to DNA. As a result, we were not able to find any ChIP-seq data for USP18.

6. For Figure 1B, as mentioned in the legend, the upper panel is described as indicating the number of trans associations detected. However, the Y-axis of the upper scatter plot is labeled as the number of genes, which is somewhat confusing. It would be helpful to clarify the meaning of each dot in the upper plot—whether it represents individual genes associated with different trans-eQTLs or some other metric?

We have clarified the caption (legend) of Figure 1B. It now reads:

(B) *Significant trans-eQTLs detected in the meta-analysis. The upper scatter plot shows the number of trans-eQTL target genes detected at each trans-eQTL locus with p-values $< 5 \times 10^{-8}$. Six trans-eQTL loci with the most target genes have been labelled with the name of the closest cis gene.*

7. For Figure 2A, the current visualization does not follow the standard format of a locuszoom plot. To improve clarity and interpretability, the authors should label the chromosome and include the surrounding genes in the region. Additionally, incorporating LD information and recombination rates for the locus would provide important context for understanding the genetic architecture and the relationships of the variants. These enhancements will make the figure more informative and align with standard practices for presenting GWAS data.

8. For Figure 2B, the authors should clearly specify the reference allele used to calculate the effect sizes. This information is crucial for interpreting the direction and magnitude of the associations presented in the figure.

We have now re-designed both Figure 2A and 2B as requested.

Here is the updated figure:

Figure 2. SLE GWAS association at the *USP18* locus is a *trans*-eQTL for interferon response genes. (A) Regional association plot for the SLE GWAS with POEMColoc imputed summary statistics and regional association plot for the lead *trans*-eQTL gene (*HERC5*) at the

USP18 locus. The *trans*-eQTL lead and GWAS lead variants (rs4819670, shown in blue) are identical and in high LD with a missense variant (rs3180408, shown in red) in the *USP18* gene. The original regional association plot for the SLE GWAS is shown on Figure S4. (B) Volcano plot of the *trans*-eQTL target genes. (C) USP18 down-regulates type I interferon signalling by restricting the access of Janus-associated kinase 1 (JAK1) to the type I interferon receptor. Illustration adapted from Alshime *et al*³⁷

9. For Figure 2C, the labeling of "type 1 interferon" should be corrected to "IFN-1" rather than "INF-I" to ensure accuracy and consistency in nomenclature. Similar corrections are needed for "INF-I receptor". Clear and precise labeling will improve the professionalism and readability of the figure.

These errors have now been fixed.

11. In line 154, the authors state that the identified USP18 *trans*-eQTL variant rs4819670 is in high LD with the missense variant rs3180408 in the USP18 gene in EUR and EAS populations. However, it is unclear whether rs4819670_C is in high LD with rs3180408_C or with rs3180408_T. Based on publicly available data from LDlink, rs4819670_C is in high LD with rs3180408_C. The authors should clarify this relationship explicitly to avoid confusion and ensure the accuracy of the interpretation regarding the functional and regulatory implications of these variants.

LD is typically measured in units of r^2 , which does not take the direction of correlation into account, but for clarity we have now also added the signed correlation coefficients to highlight that the correlation between the genotypes of these two variants is negative (due to differences in which allele was arbitrarily chosen as the reference allele).

Here is the updated text on page 5, lines 162-166:

However, the rs4819670 lead variant was in high linkage disequilibrium (LD) with a USP18 missense variant rs3180408 (chr22_18167915_C_T, ENSP00000215794.7:p.Thr169Met) in both European and East Asian populations ($r = -1$, $r^2 = 1$ in EAS and $r = -0.98$, $r^2 = 0.96$ in EUR 1000 Genomes superpopulation).

12. In lines 185-186, the authors state, "39/50 genes were also more highly expressed in peripheral blood mononuclear cells from SLE cases compared to controls (Table S4), consistent with the established role of increased interferon signaling in SLE." However, Table S4 indicates that 40 genes are labeled as highly expressed, not 39. This discrepancy should be addressed and corrected to ensure consistency between the text and the supplementary materials.

The number of differentially expressed genes has now been corrected to 40 (instead of 39) in the main text.

Reviewer #2 (Remarks on code availability):

I have checked the summary statistics. The data can be easily downloaded and include meta-analysis results.

Thank you.

Reviewer #3 (Remarks to the Author):

The author performed an extensive trans-eQTL analysis utilizing LCL samples, which lead to the identification of four robust loci. A significant discovery within this study is the link between the USP18 locus and SLE. Specifically, the SLE risk allele of rs4819670 at the USP18 locus correlates with increased expression levels of Type I interferon genes. While the study addresses an important gap in understanding the genetic variants contribution to the abnormal activation of interferon pathway, several major issues need to be addressed to strengthen the validity and impact of the findings.

Thank you.

1. How did the authors adjust for potential biases caused by the different measurement platforms—RNA-seq in some datasets and microarray in others—used in their analysis of data from various sources?

We agree with the reviewer that combining gene expression data from different quantification platforms into a single regression analysis is likely to introduce biases. This is why we performed association testing separately in each cohort and subsequently meta-analysed the results. Notably, each cohort used only one quantification method (RNA-seq or microarray, see Table S1), so the association testing was not confounded by the quantification method. We have now clarified this in the main text:

Page 3, lines 111-113: *“To avoid confounding by technical factors, we performed association testing separately in each cohort and meta-analysed the results (Figure 1A).”*

Another potential issue is that quantification methods employed by different cohorts could introduce additional heterogeneity into our meta-analysis. However, we saw that all ten loci prioritised by our primary analysis remained significant when we used a more conservative random-effect meta-analysis that explicitly models heterogeneity between cohorts (see response to point 3 from Reviewer 2).

Finally, we replicated our main discoveries in an independent RNA-seq-based cohort of 672 LCL samples, thus further strengthening the results.

2. In line 154, multiple SNPs are in linkage disequilibrium with rs4819670 across both European and East Asian populations. How do the authors ensure that the functional effect is not being driven by these other SNPs instead? Further, the authors should provide all the LD variants

information.

We have redrawn Figure 2A using LocusZoom R package and now show LD with the lead variant using the standard colour coding implemented in LocusZoom. We agree that in the absence of functional validation, we cannot completely rule out that the effect might be mediated by other variants in LD. We now explicitly acknowledge this in the main text. See response to your next point for more details.

A Imputed SLE summary statistics (POEMColoc)

C

Figure 2. SLE GWAS association at the *USP18* locus is a *trans*-eQTL for interferon response genes. (A) Regional association plot for the SLE GWAS with POEMColoc imputed summary statistics and regional association plot for the lead *trans*-eQTL gene (*HERC5*) at the *USP18* locus. The *trans*-eQTL lead and GWAS lead variants (rs4819670, shown in blue) are identical and in high LD with a missense variant (rs3180408, shown in red) in the *USP18* gene. The original regional association plot for the SLE GWAS is shown on Figure S4. **(B)** Volcano

plot of the *trans*-eQTL target genes. (C) USP18 down-regulates type I interferon signalling by restricting the access of Janus-associated kinase 1 (JAK1) to the type I interferon receptor. Illustration adapted from Alshime *et al*³⁷

3. To date, extensive research on *trans*-eQTLs has provided novel insights into the genetic underpinnings of disease onset. Nonetheless, a critical gap in this field is the shortage of functional validation. For variants located in coding regions, assessing their functionality is relatively straightforward. The study identifies rs3180408 as a potential causal variant influencing the type I IFN signaling based on statistical associations without experimental validation. To substantiate this claim, the authors should conduct functional assays, such as CRISPR-Cas9-mediated gene editing and reporter assays, to experimentally validate the impact of rs3180408 on gene expression and type I IFN signaling pathways in B cells.

We think that there are two separate questions here that can be answered with different strategies: (1) what is the causal gene at the *USP18* locus that is responsible for the observed *trans*-eQTL effects? (2) what is the exact causal variant?

For the first question, we believe that there is overwhelming evidence that the most likely causal gene is *USP18*. This includes rare diseases evidence that we highlighted already in the initial version of the manuscript (Meuwissen *et al*, 2016; Alshime *et al*, 2020). Furthermore, we now also refer to another study (Taylor *et al*. 2018, <https://doi.org/10.1002/JLB.3MIA0917-352R>) that used CRISPR/Cas9 to perform *USP18* knockout in human induced pluripotent stem cell derived macrophages and observed expected increase of a number of canonical interferon response genes. We've now included this reference in the main text (page 5, lines 169-171):

“Similarly, USP18 knock-out in human macrophages increased the expression of several canonical interferon response genes upon stimulation with interferon-beta (Taylor et al. 2018).”

For the second question, we agree with the reviewer that it would be ideal to validate the potential effect of the rs3180408 missense variant using CRISPR-Cas9-mediated gene editing. We did explore this option, but unfortunately our power calculations suggested that due to the small expected effect of the missense variant, we would need to repeat the experiment approximately 200 times to have a reasonable chance of detecting the effect experimentally. This assumes the best scenario where we are able to perfectly replace both alleles of the missense variant in an LCL cell line without introducing any additional genetic variation. Thus, we feel that such an experiment is not feasible at this stage.

To acknowledge uncertainty about the causal variant, we have now edited the main text as follows (pages 5-6, lines 174-182):

However, identifying the exact causal variant remains challenging. While the absence of cis eQTL and splicing QTL evidence suggests that the likely causal mechanism is the rs3180408 missense variant, the missense variant was predicted to be benign by all tested variant effect prediction tools available from Ensembl VEP³⁸. Alternatively, there could be other genetic

variants in the region that are not captured by current genotype imputation reference panels (such as structural variants). One potential strategy to assess the functional impact of the rs3180408 missense variant on trans-eQTL target gene expression would be genome editing, but our power calculations (see Supplementary Text) suggest that this would be extremely challenging due to the small expected effect size of the variant.

We have also removed references to the missense variant from the Abstract and the last paragraph of the introduction:

Updated Abstract:

The trans-eQTL signal at the ubiquitin specific peptidase 18 (USP18) locus colocalised with a GWAS signal for systemic lupus erythematosus (SLE). USP18 is a known negative regulator of interferon signalling and the SLE risk allele increased the expression of 50 interferon-inducible genes, suggesting that the risk allele impairs USP18's ability to effectively limit the interferon response.

Page 3, line 96:

At the USP18 locus, the trans-eQTL signal colocalised with a GWAS association for SLE.

For your reference, here is the newly added Supplementary Text:

Estimating the sample size required to experimentally validate the effect of USP18 missense variant on target gene expression

To see if it would be possible to experimentally validate the effect of the rs3180408 USP18 missense variant on trans-eQTL target gene expression, we turned to power calculations. The largest trans-eQTL effect in our discovery cohort was detected for the HERC5 gene (beta = 0.15). Given that the MAF of the rs3180408 was 0.346, this means that the rs3180408 variant explained ~1% of the variance in the expression of the HERC5 gene. However, the effect size estimate in the discovery cohort is likely to be overestimated due to the winner's curse (Huang et al. 2018). In the MAGE replication cohort (n = 682), the effect size of the rs3180408 variant was beta = 0.094, which corresponds to ~0.4% of the variance explained in the HERC5 gene expression. This is a very small effect.

To validate our simulation approach, we first sought to estimate the sample size required in a standard trans-eQTL mapping setting where the variant has a MAF = 0.346 and beta = 0.1. Averaging the results from 1000 simulations, we found that we need approximately 800 samples to have ~80% power to detect a significant effect ($p < 0.05$). This is consistent with our empirical replication results on the MAGE cohort (n = 682), where we had beta = 0.094 and $p = 0.01$.

In an ideal experimental validation scenario, we would be able to engineer this variant into a lymphoblastoid cell line in an homozygous state (i.e. change both alleles). This is expected to reduce the required sample size, because now the expected effect size is twice as large (we only compare homozygous individuals) and we can ensure that both homozygotes are present

at 50% frequency. This was confirmed in simulations where we found that we would need ~200 samples to have 80% power to detect the effect and ~300 samples to have 95% power to detect the effect.

One way to design such an experiment would be to start with 10 independent lymphoblastoid cell lines homozygous for rs3180408-C reference allele, perform genome editing to change both copies of the C allele to T allele, isolate successfully edited clones and then measure HERC5 expression in each clone and matched control cell line 10-15 times (200-300 samples total). Finally, at this scale, various technical and batch effects are likely to have a much larger effect than the (very small) genetic effect we are trying to ascertain, further complicating interpretation. Thus, even if we were able to perform genome editing in LCLs or primary B cells (which is far from obvious), the expected effect size of the missense variant is too small to be detectable in this setting.

4. In line 265, the author points out that the USP18 trans-eQTL signal is context-specificity. Compared to bulk datasets, scRNA-seq eQTL datasets are less impacted by cell composition and are therefore ideal for trans-eQTL replication, could the author perform this analysis to further support their findings?

We have now attempted to replicate the USP18 *trans*-eQTL signal in naive B cells from the OneK1K single-cell dataset (n = 844 donors). None of the 44 genes that we could test (i.e. they passed minimum gene expression level filters) were nominally associated ($p < 0.05$) with our trans-eQTL lead variant. We have modified the main text to reflect this new analysis (page 10, lines 291-295:

We further tried to replicate the USP18 trans-eQTL association in naive B cells using single-cell RNA-seq data from 844 individuals from the OneK1K cohort, but none of the tested genes were associated with the trans-eQTL lead variant (Table S6). Thus, even at these large sample sizes, the USP18 trans-eQTL signal would not have been discovered in whole blood or naive B-cells.

Minor comment:

1. In Figure 2 and Figure 3, "IFN" is misspelled as "INF", the authors should carefully check these typos.

This has now been fixed.

2. In line 162, "rs4819670 missense variant" should be "rs3180408 missense variant"?

This has now been fixed.

3. Regarding Figure 2B, the authors need to explicitly state the reference allele utilized for calculating the effect sizes.

This has now been fixed. See response to point 2 above for updated Figure 2.

Reviewer #4 (Remarks to the Author):

Reviewer #4 (Remarks on code availability):

README files are clear with enough instructions. I also checked the provided summary statistics files, and they could be downloaded and read by software. The information in the summary statistics is sufficient.

Thank you.